# Slimming the Fat-Tail: Morphing-Flow for Adaptive Time Series Modeling

**Tianyu Liu** [*1] **Kai Sun** [*2] **Fuchun Sun** [1] **Yu Luo** [3] **Yuanlong Zhang** [4]

## Abstract

Temporal sequences, even after stationarization, often exhibit **leptokurtic distributions** with fat tails and persistent distribution shifts. These properties destabilize feature dynamics, amplify model variance, and hinder model convergence in time series forecasting. To address this, we propose *Morphing-Flow (MoF)*, a framework that combines a spline-based transform layer (Flow) and a test-time-trained method (Morph), which adaptively normalizes non-stationary, fat-tailed distributions while preserving critical extreme features. MoF ensures that inputs remain within a network's effective activation space—a structured, normal-like distribution—even under distributional drift. Experiments across eight datasets show that MoF achieves state-of-the-art performance: With a simple linear backbone architecture, it matches the performance of state-of-the-art models on datasets such as Electricity and ETTh2. When paired with a patch-based Mamba architecture, MoF outperforms its closest competitor by **6.3%** on average and reduces forecasting errors in fat-tailed datasets such as Exchange by **21.7%**. Moreover, MoF acts as a plug-and-play module, boosting performance in existing models without architectural changes.

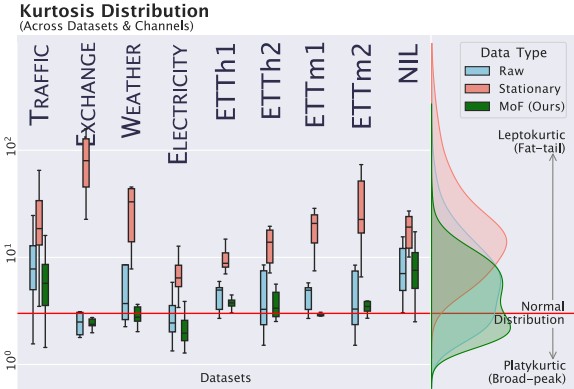

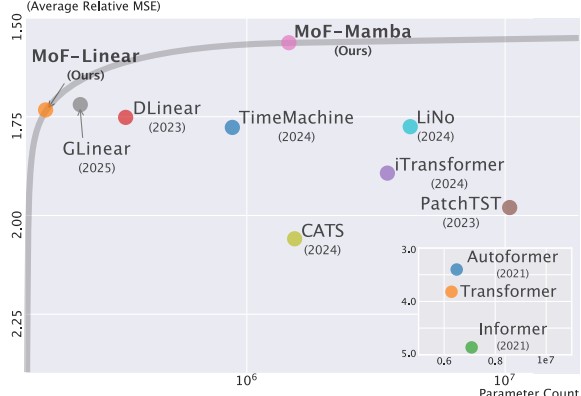

*Figure 1.* **Kurtosis Dynamics and Model Efficiency** (*Top*) Stationarization amplifies tail-heaviness, while MoF restores near-Gaussian kurtosis and stabilizes variance across channels. (*Bottom*) MoF balances simplicity and effectiveness: lightweight variants remain competitive with the baseline, and the Mamba-based backbone achieves a 6.3% performance improvement.

## 1. Introduction

Time series forecasting faces inherent challenges from non-stationary, fat-tailed distributions, phenomena prevalent in critical domains such as energy grids and financial markets, where frequent extreme events (e.g., demand surges, market

---

[*]Equal contribution [1]Department of Computer Science and Technology, Institute for Artificial Intelligence, BNRist Center, Tsinghua University, Beijing, China [2]School of Biomedical Engineering, Tsinghua University [3]Huawei Noah's Ark Lab [4]Department of Automation, Tsinghua University. Correspondence to: Fuchun Sun <fcsun@mail.tsinghua.edu.cn>.

*Proceedings of the $42^{nd}$ International Conference on Machine Learning*, Vancouver, Canada. PMLR 267, 2025. Copyright 2025 by the author(s).

shocks) deviate sharply from Gaussian assumptions. Such sequences often exhibit power-law dynamics, with fat-tailed fluctuations disproportionately influencing system behavior and destabilizing the feature distributions(Wilkinson et al., 2015). While stationarization techniques (e.g., differencing, detrending) mitigate non-stationarity, our analysis of nine benchmark datasets reveals a critical oversight: stationarization often amplifies leptokurtic behavior (Fig 1), exacerbating tail risk and inflating prediction variance. Empirical studies further indicate that fat-tailed, leptokurtic noise (Simsekli et al., 2019; Nguyen et al., 2019) violates the assumption of bounded variance, complicating optimiza-

tion (Zhang et al., 2020; Wang et al., 2021). This presents a fundamental dilemma: aggressive constraints risk erasing critical extremal values (essential for predicting disasters or market shifts), while retaining raw distributions introduces unstable gradients and divergent training dynamics.

Existing approaches have difficulty addressing this tension. Non-differentiable operations (e.g., clipping), static parametrizations, and Gaussian-centric frameworks (e.g., Box-Cox transformations) often fail to adequately address dynamic distribution shifts and fat-tailed phenomena (Chavez-Demoulin & Davison, 2012). While gradient clipping (Gorbunov et al., 2020; Yang et al., 2022) and decomposition architectures (RB, 1990; Hamilton, 2020) partially mitigate non-stationarity, their robustness to pervasive leptokurtic noise remains insufficient. This underscores the need for architectures that explicitly adapt to fat-tailed distributions while balancing theoretical rigor with practical resilience.

We propose Morphing-Flow (MoF), a framework designed to dynamically map input distributions to a well-structured, normal-like space within neural activation regions, preserving extremal features and stabilizing training. MoF integrates two components:

- **Flow**: A monotonic spline transformation that learns channel-specific mappings via piecewise polynomial functions, thereby reducing kurtosis while ensuring invertibility and differentiability.
- **Morph**: A lightweight test-time adaptation layer that iteratively adjusts Flow's parameters to counteract instance-level distribution shifts and enhance robustness to temporal drift.

The fully invertible architecture of the MoF enables end-to-end training without auxiliary losses, while its spline-based design adaptively balances local trends and global extremes. With a linear backbone, it matches state-of-the-art performance despite its simplicity. By mitigating fat-tailed behavior in the output distribution, MoF implicitly stabilizes gradient dynamics, a property that becomes increasingly beneficial in deeper architectures. When integrated with a patch-based Mamba backbone, MoF outperforms the closest competitor by 6.3% on average, achieving 21.7% gains on datasets with pronounced heavy-tailed behavior (e.g., *Exchange-Rate*). The MOF operates as a plug-and-play module, requiring no architectural modifications to existing models.

## 2. Impact of Fat-Tails on Convergence

**(How) Do fat-tail impede model convergence?** To investigate the impact of fat-tailed distributions on neural network training, we design a synthetic forecasting task: predicting

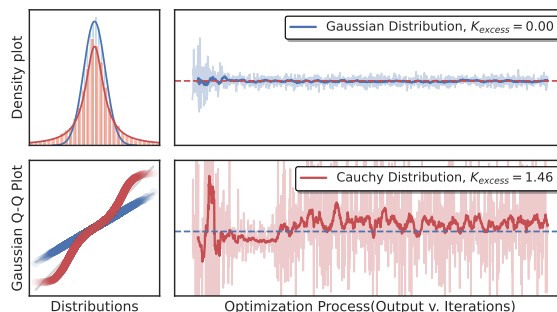

*Figure 2.* **Impact of Fat-Tailed Distributions on Convergence. Top left:** A comparison of the Probability Density Functions (PDFs) between the Gaussian (Blue, $K_{\text{Excess}} = 0$) and clipped Cauchy (Red, $K_{\text{Excess}} \approx 1.5$) distributions. **Bottom left:** A Q-Q plot highlights the heavier tails of the Cauchy distribution. **Right:** Training convergence results. The Gaussian distribution rapidly converges to the correct value (indicated by the horizontal dashed line), while the sequence sampled from the Cauchy distribution converges to an incorrect value before diverging and exhibiting oscillations.

the first moment (the mean) of a non-stationary time series. Consider a time series $\{x_t\}_{t=1}^n$, where:

$$x_t = \mu_t + \epsilon_t, \quad \mu_t = k\mu_{t-1},$$

with $k > 1$, a constant growth factor, $\mu_t$ representing the trend, and $\epsilon_t$ denoting noise drawn from a zero-mean distribution with variance $\sigma^2$. We train a linear model $f_\theta$ to predict $x_{n+1}$ from the historical observations $x_1, \ldots, x_n$,

$$\hat{x}_{n+1} = f_\theta(x_1, \ldots, x_n).$$

To compare the effects of heavy tails, we sample $\epsilon_t$ from two distributions: (1) a Gaussian $\mathcal{N}(0, \sigma^2)$ and (2) a clipped Cauchy distribution (bounded within $[-6, 6]$ to satisfy the finite-moment requirements of stochastic gradient descent), whose *excess kurtosis*

$$K_{\text{excess}} = \frac{\mathbb{E}\big[(X - \mu)^4\big]}{\sigma^4} - 3 \approx 1.5.$$

In addition, we employ a progressively diminishing learning rate to ensure training stability in the presence of these fat-tailed distributions.

Visualization of the training process is shown in Figure 2. The fat-tailed nature of Cauchy distributions (evident in the Q-Q plots and PDFs of Figure 2) has a significant impact on the training dynamics. Although Gaussian noise does not hinder stable convergence, clipped Cauchy noise with moderate excess kurtosis ($K_{\text{Excess}} = 1.46$) causes oscillations, slow convergence, or divergence. Extreme deviations in the tails of the Cauchy distribution initially bias models toward incorrect solutions, thereby exacerbating optimization

instability. Additional results reveal increased sensitivity to initialization and learning rates compared to Gaussian regimes, consistent with observations and theoretical analysis (Appendix B.2).

To learn high-variance features without causing instability, we propose an adaptive transformation module. It transforms fat-tailed distributions (e.g., clipped Cauchy) into stable, normally distributed activations, while also reducing gradient noise. By preserving training-friendly distributions in transformed spaces, models can leverage fat-tailed patterns while avoiding the high-variance noise associated with such features.

## 3. Methodology

In this section, we introduce the proposed MoF framework, illustrated in Fig. 3. We first define the problem and then detail the Flow layer for fat-tailed feature distributions and the Morph layer to address distribution shifts.

**Problem Formulation and Notations** We address multivariate time series forecasting: given $X \in \mathbb{R}^{C \times T}$ where $C$ denotes the number of channels and $T$ the number of time steps, the goal is to predict the next $F$ steps $\widehat{Y} \in \mathbb{R}^{C \times F}$. The aim is to develop a predictor such that $\widehat{Y}$ closely approximates the ground-truth observations $Y \in \mathbb{R}^{C \times F}$.

**Overall Framework** The *MoF* framework combines two synergistic components: the *Flow* layer $F(\mathbf{x} \mid \mathbf{W}_{\text{flow}})$ that mitigates fat-tailed feature distributions via learnable reshaping, and the *Morph* module, a lightweight test-time adaptation mechanism that addresses temporal distribution shifts through $\mathbf{x}_{\text{mod}} = \text{Morph}(F(\mathbf{x}))$. The Flow layer preserves the fidelity of training distributions through structured transformations, while the Morph module dynamically adapts to evolving patterns during deployment, collectively ensuring robustness to distribution shifts while maintaining operational stability.

Since both the input and output spaces exhibit fat-tailed effects, the MoF model is designed to be reversible. To achieve this, we introduce both MoF and MoF$^{-1}$ before and after the backbone, enabling the backbone to operate in a well-defined environment resilient to end-to-end distribution shifts.

### 3.1. Flow: Spline-Based Reversible Transform Layer

The residual component $x_{\text{res}}$ often exhibits complex relationships and elevated variance, characterized by a fat-tailed distribution that can impede model convergence.

To address this issue, we propose a strictly monotonic and invertible transformation, denoted as $F(x \mid \mathbf{W}_{\text{flow}})$, to map features within the bounded domain $[-T, T]$ to a new domain with the same bounds. Here, $T$ represents the do-

main's limit, and features outside this range are retained. To ensure computational efficiency and provide an explicit closed-form solution for the first-order derivatives in both the forward and reverse directions, we construct a vectorized piecewise-linear spline transformation. Let $C$ denote the number of channels and $B$ the number of bins per channel, a hyperparameter that determines the discretization of the space. For each channel $c \in \{1, \ldots, C\}$, the learnable parameters $\{w_{c,i}\}_{i=1}^{B}$ and $\{h_{c,i}\}_{i=1}^{B}$, which define the bin widths and heights respectively, are collectively denoted as $\mathbf{W}_{\text{flow}}$.

The model computes normalized bin widths $W_{c,i}$ and heights $H_{c,i}$ by applying a softplus activation function, ensuring that their respective sums each equal $2T$. The knot positions $x_{c,i}$ and $y_{c,i}$ are obtained by taking the cumulative sums of $W_{c,i}$ and $H_{c,i}$. The slope of each bin is precomputed as:

$$s_{c,i} = \frac{y_{c,i+1} - y_{c,i}}{x_{c,i+1} - x_{c,i}}. \tag{1}$$

The forward transformation maps an input $x \in [-T, T]$ to an output $y \in [-T, T]$ using the following linear relationship:

$$y = y_{c,i} + s_{c,i} \cdot (x - x_{c,i}), \tag{2}$$

where the appropriate bin $i$ is determined by identifying the interval containing $x$.

The inverse transformation involves first locating the corresponding bin for $y$ and applying the inverse linear relationship through the relation:

$$x = x_{c,i} + \frac{1}{s_{c,i}} \cdot (y - y_{c,i}). \tag{3}$$

One such transformation is defined for each input channel. This formulation ensures efficient, differentiable, and vectorized forward and inverse transformations, making it suitable for applications requiring continuous and invertible mappings.

### 3.2. Morph: Test-Time Trained Adaptive Spline Modification

Real-world time series often exhibit *distribution shift*, where the data distribution varies both across training batches and between the training and testing datasets. Despite the representational flexibility of the Flow transformation $F(x \mid \mathbf{W}_{\text{flow}})$, a fixed $\mathbf{W}_{\text{flow}}$ may fail to adapt to the shifting distributions between training and testing sets.

To address this issue, we propose a *Morph* module that modifies $\mathbf{W}_{\text{flow}}$ at test time based on the Flow-transformed sequence $F(x \mid \mathbf{W}_{\text{flow}})$, enabling the module to handle distribution shifts effectively. We first apply the vectorized

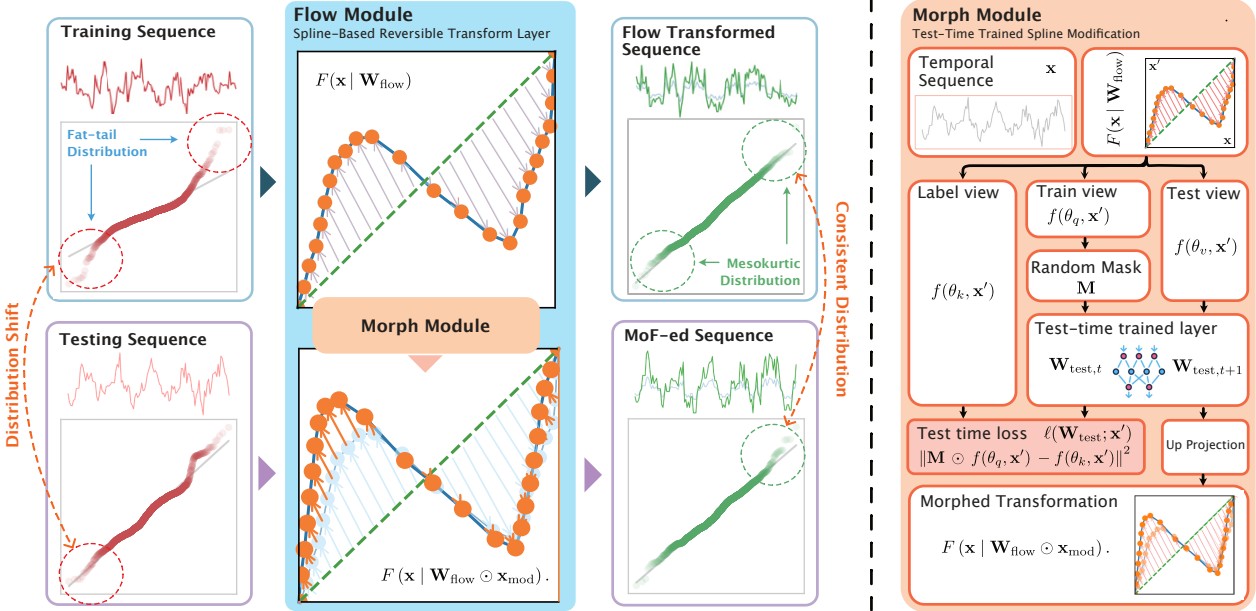

*Figure 3.* **MoF Framework**. Gaussian Q-Q plots (ETTh2) demonstrate heavy-tailed training/test (red) distributions. *Flow* transforms these into a normalized latent space (green), reducing tail effects while preserving critical features. *Morph* corrects instance/train-test shifts through adaptation, ensuring consistent distributions resistant to non-stationarity.

piecewise-linear spline (Flow Layer) as follows:

$$\mathbf{x}' = F\Big(\mathbf{x} \ \Big| \ \mathbf{W}_{\text{flow}} = \{w_{c,i}, h_{c,i}\}_{i=1}^{B}\Big), \qquad (4)$$

where $w_{c,i}$ and $h_{c,i}$ denote the original bin widths and heights, respectively.

**Test-Time Trained Temporal Layer.** Inspired by (Sun et al., 2024), we introduce a test-time-trained temporal layer designed to adapt the parameters of $\mathbf{W}_{\text{flow}}$ during inference.

We begin by constructing a self-supervised task to optimize $\mathbf{W}_{\text{test}}$ during test time. Specifically, we define low-rank projections and binary masks as follows:

$$\begin{aligned} \mathbf{x}_q &= \mathbf{M} \odot f(\theta_q, \mathbf{x}')\mathbf{W}_{\text{test}}^{\top}, \\ \mathbf{x}_k &= f(\theta_k, \mathbf{x}'), \end{aligned} \qquad (5)$$

where $f(\theta, \mathbf{x}')$ represents a learned linear transformation parameterized by $\theta$, and $\mathbf{M} \sim \text{Bernoulli}(p)$ is a binary mask.

A self-supervised loss function $\ell(\mathbf{W}_{\text{test}}; \mathbf{x}')$ is computed over batches during test time to stabilize gradient flow:

$$\ell(\mathbf{W}_{\text{test}}; \mathbf{x}') = \|\mathbf{M} \odot f(\theta_q, \mathbf{x}') - f(\theta_k, \mathbf{x}')\|^2. \qquad (6)$$

During inference, we iteratively update the test-time weight matrix $\mathbf{W}_{\text{test}}$ after each forward pass or multiple passes:

$$\mathbf{W}_{\text{test},t+1} = \mathbf{W}_{\text{test},t} - \eta \, \nabla_{\mathbf{W}_{\text{test},t}} \, \ell(\mathbf{W}_{\text{test},t}; \mathbf{x}'), \qquad (7)$$

where $\eta$ is a learnable step size enabling adaptive learning during test-time updates.

Importantly, only the matrix $\mathbf{W}_{\text{test}}$ is updated during this process, while all other parameters ($\theta_q, \theta_k, \theta_v$) remain fixed from the training phase.

**Up-Projection and Morphing.** We introduce a scaling matrix $\mathbf{x}_{\text{mod}} \in \mathbb{R}^{B \times 2}$ using the updated weight matrix $\mathbf{W}_{\text{test},t+1}$ and an *Up-Projection* layer, which is formally defined as:

$$\mathbf{x}_v = f(\theta_v, \mathbf{x}') \, \mathbf{W}_{\text{test},t+1}^{\top}, \qquad (8)$$

$$\mathbf{x}_{\text{mod}} = \text{UpProj}(\mathbf{x}_v), \qquad (9)$$

where $\text{UpProj}(\cdot)$ denotes a learned linear mapping that maps the low-dimensional embedding into a $B \times 2$ matrix. This transformation enables adaptation of the feature space to a higher-dimensional space.

Subsequently, we apply element-wise multiplication of $\mathbf{W}_{\text{flow}}$ with $\mathbf{x}_{\text{mod}}$ to adaptively morph the Flow parameters and obtain the MoF-ed feature:

$$\mathbf{x}' = F\Big(\mathbf{x} \ \Big| \ \mathbf{W}_{\text{flow}} \odot \mathbf{x}_{\text{mod}}\Big). \qquad (10)$$

### 3.3. Model Backbones

A prediction backbone is constructed using the MoF output, with its predictions fed into $\text{MoF}^{-1}$. The reversible design of MoF ensures that the backbone operates within

---
**Algorithm 1** MoF Module Pseudocode

---
**Input:** Input sequence $\mathbf{x}$ of shape $[C, T]$
**Output:** Transformed sequence $\mathbf{x}'$ of shape $[C, T]$
Apply the Flow transformation to the input sequence:
$\quad\quad \mathbf{x}' = F(\mathbf{x} \mid \mathbf{W}_{\text{flow}})$
Feed $\mathbf{x}_{\text{flow}}$ into the temporal morph module:
$\quad\quad \mathbf{x}_{\text{mod}} = \text{Morph}(\mathbf{x}_{\text{flow}})$
$\quad\quad$ // $x_{mod}$ has shape $B \times 2$
Transform the sequence using the modified parameters:
$\quad\quad \mathbf{x}' = F(\mathbf{x} \mid \mathbf{W}_{\text{flow}} \odot \mathbf{x}_{\text{mod}})$
**Return:** $\mathbf{x}'$

---

its intended kurtosis range. Importantly, the MoF module is trained end-to-end without auxiliary losses to constrain its structure. To effectively demonstrate the performance of the MoF module, we design two simple and minimal backbones.

**MoF Linear:** A remarkably simple linear backbone directly regresses historical time series to predict future values using a weighted sum operation ($\hat{\mathbf{x}} = W\mathbf{x} + b$), as in LTSF-Linear (Zeng et al., 2023). To demonstrate our module's capability, we omit trend decomposition and apply **instance-wise z-score normalization** for stabilization, which standardizes residuals by removing instance-specific means and variances.

**MoF Mamba** Adopting the patching methodology of PatchTST (Nie et al., 2023), we segment the inputs into $P$-length patches with a stride of $S$, reducing the number of tokens to $\lceil L/S \rceil$ after padding. The Mamba backbone (Gu & Dao, 2023) processes these patches without positional embeddings using state-space dynamics, fully replacing traditional Transformers. For long-term stability, we decompose the inputs using **moving-average kernels**, linearly predict the trend component and processing residuals with Mamba. Predictions are generated through flattening and a linear head, and combined with the trend predictions.

## 4. Experiments

We evaluate the proposed model through comprehensive experiments: Section 4.1 benchmarks MoF Linear and MoF Mamba against nine state-of-the-art (SOTA) methods across eight datasets; Section 4.2 reports performance improvements, highlighting robustness against fat-tailed noise and non-stationarity; Section 4.3 explores module contributions via ablation studies, isolating the effects of Flow's normalization and Morph's adaptation; and Section 4.4 investigates hyperparameter sensitivity, efficiency trade-offs, and generalization performance when integrated with diverse backbones.

### 4.1. Experimental Settings

**Datasets:** We conduct experiments on eight widely used multivariate time series forecasting datasets, including Electricity Transformer Temperature (ETTh1, ETTh2, ETTm1, and ETTm2) (Zhou et al., 2021), Electricity, Traffic, Weather (Wu et al., 2021), and Exchange Rate. To ensure consistency, we follow the standard protocol (Liu et al., 2023) and split the datasets into training, validation, and test sets with a 6:2:2 ratio for ETT datasets and a 7:1:2 ratio for the remaining datasets. Detailed dataset characteristics are provided in Appendix C.1.

**Evaluation Protocol:** Following the standard Autoformer protocol (Wu et al., 2021), we adopt Mean Squared Error (MSE) and Mean Absolute Error (MAE) as evaluation metrics. Due to observed variability in test results, each experiment is repeated five times, and Mann-Whitney U tests are used to evaluate statistical significance of performance differences. The historical horizon length is set to $T = 336$, with prediction lengths $F \in \{96, 192, 336, 720\}$. All hyperparameters are held constant across datasets to avoid the confounding effects of dataset-specific tuning, which might otherwise lead to overly optimistic results that fail to generalize. This unified setting may result in minor discrepancies from original baseline reports; nevertheless, as shown in Appendix D.4, our reproduced baselines remain competitive and consistent across benchmarks. Detailed hyperparameters for MoF Linear and MoF Mamba are included in Appendix D.

**Baseline Models:** We compare MoF against nine representative models, spanning both recent state-of-the-art methods and established baselines. Recent models include GLinear (Rizvi et al., 2025), LiNo (Yu et al., 2024a), TimeMachine (Ahamed & Cheng, 2024), CATS (Kim et al., 2024), iTransformer (Liu et al., 2024a), and PatchTST (Nie et al., 2023). Widely acknowledged baselines include DLinear (Zeng et al., 2023), Informer (Zhou et al., 2021), and Autoformer (Wu et al., 2021). These methods span Transformer-based, Linear-based, and Mamba-based architectures, enabling a comprehensive evaluation.

### 4.2. Experimental Results

**MoF-based Model Improves Over State-of-the-Art.** The experimental results are summarized in Table 1, with detailed results for different prediction horizons and statistical significance tests (Mann-Whitney U tests) provided in Appendix E. The MoF-based model consistently outperforms all competitors across all 8 datasets. On average, it surpasses the next best competitor by 6.3% and achieves substantial improvements over widely used models such as iTransformer and PatchTST, with gains of 14.9% and 16.9%, respectively. Furthermore, our MoF-Mamba model outperforms recent models, including TimeMachine and

*Table 1.* Performance comparison of our proposed model (MoF) with Mamba and Linear backbones against baseline methods for multivariate long-term forecasting. Results are averaged **across 4 horizons** and represent the **mean MSE from 5 independent runs** using different random seeds. The best results are highlighted in **bold**, while the second-best results are underlined. Model names ending with "former" are omitted for brevity. "TimeM." denotes TimeMachine. Detailed results, including extended comparisons with methods like IN-Flow and FITS, are provided in Appendix E and Appendix K.

| DATASET | MEDIAN $K_{excess}$ | MoF (MAMBA) | MoF (LINEAR) | GLINEAR (2025) | LiNo (2024B) | TimeM. (2024) | CATS (2024) | iTRANS. (2023) | PATCHTST (2023) | DLINEAR (2023) | AUTO. (2021) | IN. (2021) | TRANS. (2017) |
|---|---|---|---|---|---|---|---|---|---|---|---|---|---|
| ETTh1 | 5.80 | **0.501** | 0.515 | 0.522 | 0.528 | 0.529 | 0.822 | 0.538 | 0.584 | 0.524 | 0.771 | 1.243 | 0.828 |
| ETTh2 | 10.9 | 0.224 | **0.222** | 0.240 | 0.248 | 0.241 | 0.263 | 0.261 | 0.263 | 0.228 | 0.374 | 0.458 | 0.431 |
| ETTm1 | 17.7 | **0.392** | 0.401 | 0.395 | 0.407 | 0.403 | 0.414 | 0.442 | 0.416 | 0.452 | 0.619 | 0.745 | 0.655 |
| ETTm2 | 19.6 | **0.165** | 0.183 | 0.166 | 0.166 | 0.165 | 0.172 | 0.182 | 0.169 | 0.170 | 0.199 | 0.381 | 0.293 |
| ELECTRICITY | 3.42 | **0.163** | 0.164 | 0.167 | 0.175 | 0.166 | 0.221 | 0.165 | 0.164 | 0.167 | 0.221 | 0.354 | 0.291 |
| EXCHANGE | 74.2 | **0.305** | 0.402 | 0.394 | 0.403 | 0.432 | 0.397 | 0.462 | 0.544 | 0.390 | 1.198 | 1.676 | 1.224 |
| TRAFFIC | 15.6 | **0.416** | 0.430 | 0.428 | 0.458 | 0.417 | 0.937 | 0.451 | **0.416** | 0.435 | 0.648 | 0.811 | 0.691 |
| WEATHER | 30.0 | **0.221** | 0.222 | 0.226 | 0.238 | 0.227 | 0.225 | 0.241 | 0.234 | 0.245 | 0.361 | 0.529 | 0.505 |
| 1ST. COUNT | | 43/64 | | 6 | 0 | 1 | 0 | 0 | 9 | 5 | 0 | 0 | 0 |
| 2ND. COUNT | | 27/64 | | 7 | 3 | 10 | 4 | 1 | 5 | 7 | 0 | 0 | 0 |
| AVERAGE MSE | | **0.298** | 0.317 | 0.317 | 0.328 | 0.323 | 0.431 | 0.343 | 0.349 | 0.326 | 0.549 | 0.775 | 0.615 |
| DIFFERENCE | | **0.0%** | -6.4% | -6.3% | -9.9% | -8.1% | -44.6% | -14.9% | -16.9% | -9.4% | -84.0% | -160% | -106% |

LiNo, by 8.1%–9.9%. Notably, on the dataset with the highest kurtosis (Exchange), our model outperforms the closest competitor by a margin of 21.7%.

Additionally, we incorporate IN-Flow and FITS in our extended analysis (Appendix K) to provide a broader perspective on normalization-centric approaches. While IN-Flow also builds on flow-based transformations, it focuses on using entangled flows to model non-stationarity—a fundamentally different objective from our focus on stabilizing fat-tailed gradients. Despite this distinction, we include IN-Flow in the comparison due to its similar building blocks. We observe that IN-Flow performs reasonably on smaller datasets but struggles with high-dimensional inputs. FITS, on the other hand, prioritizes parameter efficiency through adaptive feature reweighting, though it is sensitive to hyperparameter settings and exhibits instability under our unified evaluation protocol.

**MoF-based Model Excels in Long-Range Prediction.** As shown in Fig. 4, the MoF-based model achieves superior long-term forecasting performance, significantly outperforming competitors by **16.5%** in relative MSE (as defined by the ratio to the best-performing baseline across all horizons for each dataset). This improvement stems from MoF's ability to mitigate fat-tailed distributions and stabilize sporadic, high-variance signals: the Flow module aligns residuals to a near-Gaussian latent space (evidenced by kurtosis reduction in Fig. 1), reducing outlier-driven variance while preserving extremal patterns, and the Morph module dynamically adapts to distributional shifts during inference, ensuring stable convergence under non-stationarity. By jointly addressing fat-tailed noise and temporal instability, MoF enhances the backbone's capacity to model long-range dependencies—critical for minimizing error compounding over extended horizons.

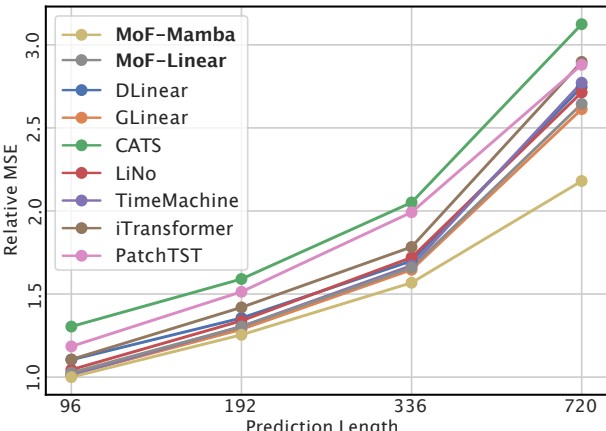

*Figure 4.* Relative Mean Squared Error (MSE) across varying time horizons, averaged over eight datasets. The relative MSE is calculated as the average ratio of the MSE relative to the best-performing model across all prediction lengths and configurations.

### 4.3. Modular Analysis

**What the Spline (Flow) Learned.** Figure 5 illustrates MoF's distribution transformation across three phases: (1) non-stationary raw inputs with fat tails, (2) stationarized features that retain excess kurtosis ($K_{\text{excess}} > 29$), and (3) outputs approaching quasi-normality ($K_{\text{excess}} < 1$). MoF systematically reduces channel-wise excess kurtosis by 18.5%–93.3% (Appendix F). Its reversible architecture ensures optimization-friendly distributions at both model inputs (historical data) and outputs (future predictions), thereby providing the backbone with a stable and well-structured feature space.

**Gradient-level stability.** Figure 6 illustrates the evolution of gradient variance during training on the *ETTm1* dataset, comparing instance-wise $z$-score, RevIN, and our proposed MoF. MoF consistently demonstrates lower and more stable

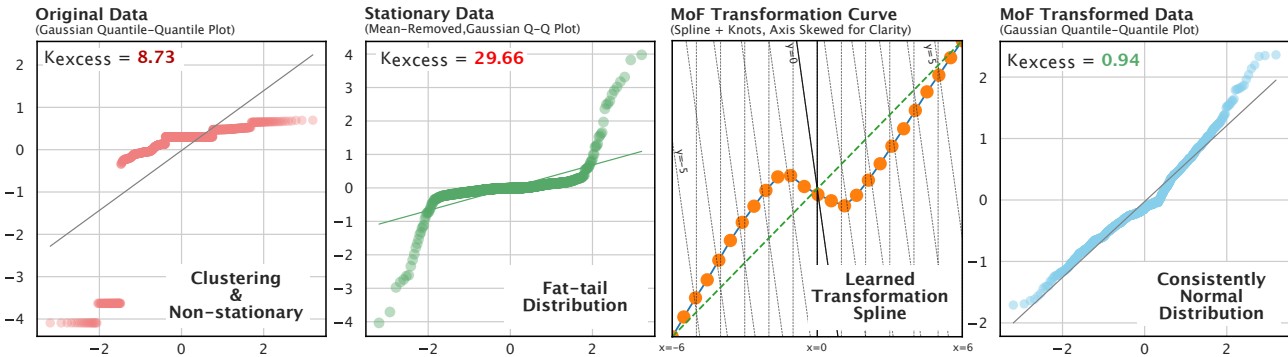

*Figure 5.* Transformation process visualization (ETTh2 dataset, single channel): (1) Raw non-stationary data exhibits non-Gaussian characteristics in the Q-Q plot against a normal distribution, with pronounced deviation from linearity; (2) The stationarized sequence is centered at the origin but retains fat tails (excess kurtosis $K_{\text{excess}} = +29.66$); (3) The learned MoF transformation curve obtained via end-to-end training, with vertical axis scaling adjusted for spline visualization; (4) The final transformed distribution approximates normality with excess kurtosis $K_{\text{excess}} = +0.94$ after Flow processing.

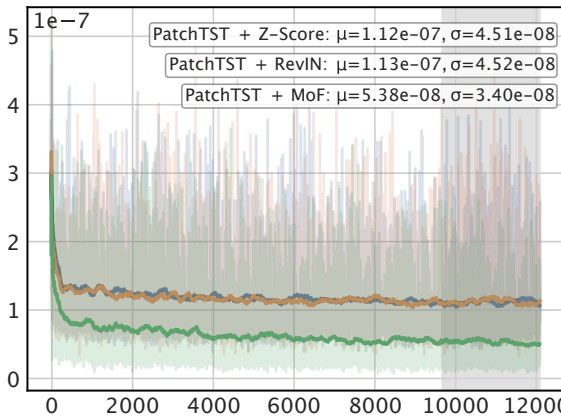

*Figure 6.* Gradient variance across training iterations on ETTm1 under different normalization strategies is shown. MoF exhibits lower variance and greater stability. Statistics (mean $\mu$, standard deviation $\delta$) of grad-var are computed over the last 2000 iterations.

gradient variance throughout training. The summary statistics (mean and standard deviation) are computed over the final 2000 iterations to characterize steady-state behavior.

This trend is more pronounced in deeper models (e.g., PatchTST), particularly on heavy-tailed datasets such as *Exchange*. In addition to variance, we track gradient norm, skewness, and (excess) kurtosis—where MoF again shows improved numerical conditioning and smoother training dynamics. These gradient-level benefits correspond to the observed gains in accuracy under fat-tailed noise. Full statistics and additional visualizations are available in Appendix M.

**Cross-Channel Consistency** Furthermore, MoF achieves a 97.27% reduction in inter-channel kurtosis variance (Appendix F). This variance reduction directly stabilizes training by aligning the magnitudes of gradients across channels, a step critical for preventing divergent learning trajectories in multi-series forecasting. The spline-learned, channel-consistent normalizing flow retains the expressiveness of

temporal patterns while enabling stable gradient propagation. Notably, this channel-aligned stabilization is essential for modern channel-independent (CI) architectures, where reduced distributional disparities enhance the effectiveness of shared parameters across individual time series.

**Ablation Study of the Test-Time Training Module:** While the experiments above demonstrate the effectiveness of the proposed modules, further analysis reveals that the impact of the TTT module is context-dependent. In summary, it is more beneficial in the short term but comes with minor drawbacks in the long term. As shown in Figure 7, we conducted an ablation study of the TTT module across various datasets on an expanded range of prediction lengths. The results indicate that TTT significantly improves performance in shorter-term prediction windows (up to a 5% error reduction). However, for longer-term predictions, the benefits of TTT are less pronounced, as these tasks are inherently more challenging due to higher noise levels and reduced sensitivity to distributional shifts.

### Superiority of MoF over other normalization methods

Although our method (MoF) primarily focuses on addressing skewness and kurtosis in data distributions, it can structurally and functionally complement existing approaches targeting non-stationarity. We compared MoF with FAN (Ye et al., 2024), RevIN (Kim et al., 2021), SAN (Liu et al., 2024b), and DishTS (Fan et al., 2023) using DLinear as the backbone, with the input horizon fixed at 336. Evaluations were conducted on 8 datasets across 4 prediction lengths. To demonstrate the plug-and-play capability of MoF, all modules were tested using default parameters without requiring dataset-specific tuning.

The experimental results are shown in Fig. 8, with detailed quantitative results provided in Appendix I. The results demonstrate that even when using basic instance-wise z-score normalization, MoF outperforms other models de-

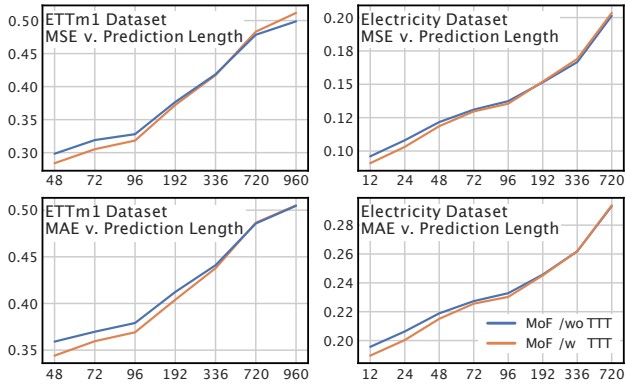

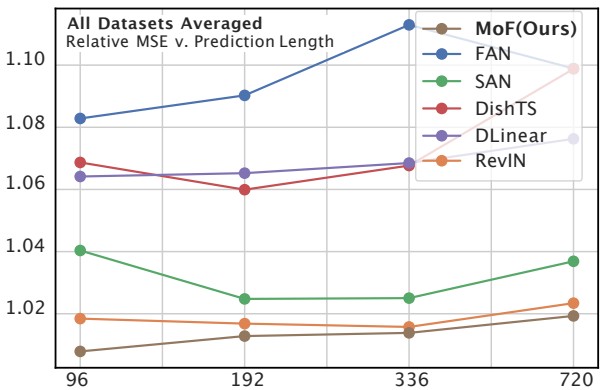

*Figure 7.* Ablation study of the Test-Time Training (TTT) module across different datasets. TTT significantly improves short-term performance (up to 5% error reduction) while showing less pronounced benefits for longer-term predictions, where distributional shifts are relatively less impactful.

*Table 2.* Hyperparameter Sensitivity Analysis of MoF

| | ETTH1 | | ETTM1 | | WEATHER | |
|---|---|---|---|---|---|---|
| | MSE | MAE | MSE | MAE | MSE | MAE |
| BIN | | | TAIL = 6.0 | | | |
| 6 | 0.4688 | 0.4806 | 0.3802 | 0.4165 | **0.1918** | **0.2416** |
| 12 | **0.4678** | 0.4797 | 0.3780 | 0.4162 | 0.1952 | 0.2441 |
| 24 | 0.4685 | **0.4795** | 0.3778 | 0.4158 | 0.1932 | 0.2441 |
| 48 | 0.4689 | 0.4798 | **0.3777** | **0.4136** | 0.1944 | 0.2448 |
| **DIFF** | 0.23% | 0.67% | 1.73% | 0.22% | 0.71% | 1.32% |
| TAIL | | | BIN COUNT = 24 | | | |
| 3.0 | 0.4688 | 0.4798 | **0.3764** | **0.4112** | 0.1927 | 0.2463 |
| 6.0 | 0.4672 | 0.4793 | 0.3770 | 0.4144 | 0.1927 | 0.2459 |
| 8.0 | 0.4673 | 0.4794 | 0.3787 | 0.4164 | 0.1939 | 0.2435 |
| 12.0 | **0.4622** | **0.4768** | 0.3792 | 0.4182 | 0.1961 | 0.2435 |
| 18.0 | 0.4672 | 0.4801 | 0.3903 | 0.4232 | 0.1917 | 0.2419 |
| 24.0 | 0.4656 | 0.4789 | 0.3855 | 0.4204 | **0.1902** | **0.2402** |
| **DIFF** | 1.40% | 3.54% | 3.02% | 0.69% | 2.83% | 2.48% |

of 3.54%. These findings confirm MoF's robustness and plug-and-play flexibility across different datasets.

**Model Efficiency** MoF achieves parameter-efficient performance, as shown in Fig. 1. Linear-MoF attains 8.5% higher accuracy than iTransformer with $21\times$ fewer parameters. Against TimeMachine, Linear-MoF improves performance by 1.7% using $5.28\times$ fewer parameters, while Mamba-MoF outperforms it by 8.1% at identical parameter scales (Appendix G).

This efficiency originates from MoF's adaptive feature transformation, which confines activations to stable operational ranges while mitigating training instability caused by fat-tailed, high-variance inputs. The design eliminates redundant parameters typically required to handle distributional extremes.

**Complexity Analysis** The computational complexity of the proposed MoF framework consists of two components: the Flow layer and the Morph module. The Flow layer operates with complexity $\mathcal{O}(C\,T\,B)$, where $C$ is the number of input channels, $T$ is the sequence length, and $B$ is the number of spline bins per channel. The Morph module introduces an additional overhead of $\mathcal{O}(N_{\mathrm{iter}}\,C\,T\,d + d\,B + C\,T\,B)$, where $d$ is the internal projection dimension and $N_{\mathrm{iter}}$ is the number of gradient updates performed at test time. The three terms respectively account for test-time adaptation, projection to Flow parameters, and a second application of the Flow transformation. Since $d \ll C\,T$, $B$ is moderate, and $N_{\mathrm{iter}}$ is small (typically 1–5), the overall asymptotic overhead introduced by Morph remains bounded. Empirically, Morph contributes only **17.92%** of MoF's inference time on average, with less than **5%** increase in total runtime over Flow-only baselines. A detailed complexity analysis and wall-clock profiling across six datasets are provided in Appendix L.

*Figure 8.* Relative MAE comparison of MoF and other normalization methods across all 8 datasets and 4 prediction horizons. Relative MSE is calculated w.r.t. the best-performing model for each prediction length.

signed for non-stationary distributions, highlighting the substantial impact of fat-tailed distributions.

### 4.4. Model Analysis

**Hyperparameter Robustness of MoF Module** The MoF module introduces two hyperparameters: Tail-size ($T$) and Bin Count ($B$). Our experiments demonstrate that $T$ and $B$ require no dataset-specific tuning and have limited impact on performance and convergence. Previous experiments fixed $T = 6.0$ and $B = 24$. Increasing $B$ allows finer feature representation but linearly increases parameter size, while a larger $T$ enhances resilience to high-variance noise at the cost of reduced parameter density in the core feature space.

Table 2 shows the sensitivity analysis: fixing $T = 6.0$ and varying $B$ results in a fluctuation of up to 1.73%, whereas fixing $B = 24$ and varying $T$ yields a maximum difference

*Table 3.* Comparison of model performance with and without MoF module across various datasets.

| MODEL | AUTOFORMER | | | INFORMER | | | DLINEAR | | |
|---|---|---|---|---|---|---|---|---|---|
| | RAW | +MoF | DIFF. | RAW | +MoF | DIFF. | RAW | +MoF | DIFF. |
| ETT(ALL) | 0.491 | **0.455** | **-5.64%** | 0.707 | **0.508** | **-28.1%** | 0.344 | **0.330** | **-3.96%** |
| ELECT. | 0.220 | **0.200** | **-9.05%** | 0.354 | **0.235** | **-33.8%** | 0.167 | **0.164** | **-1.59%** |
| EXCHG. | 1.198 | **0.690** | **-29.3%** | 1.676 | **0.694** | **-58.6%** | **0.390** | 0.402 | 3.16% |
| TRAFFIC | 0.648 | **0.677** | **-5.55%** | 0.812 | **0.742** | **-8.63%** | 0.435 | **0.430** | **-0.96%** |
| WEATHER | 0.361 | **0.313** | **-18.4%** | 0.529 | **0.356** | **-32.7%** | 0.245 | **0.222** | **-9.15%** |
| AVERAGE | 0.584 | **0.467** | **-20.0%** | 0.816 | **0.507** | **-37.8%** | 0.316 | **0.310** | **-1.93%** |

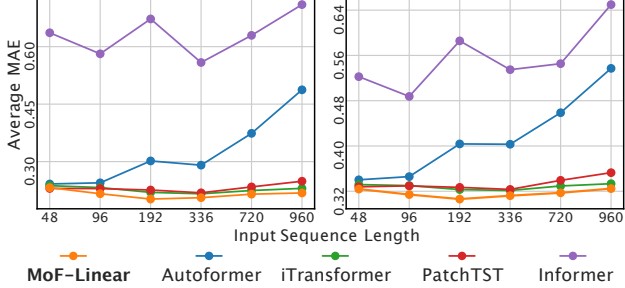

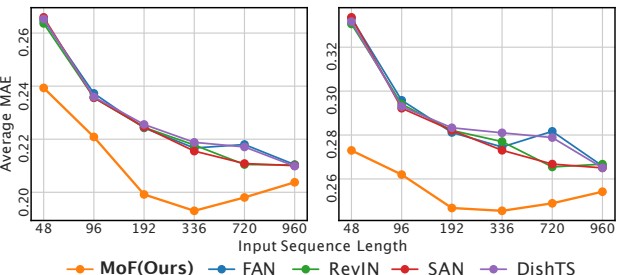

*Figure 9.* Input-length sensitivity on ETTH2, showing non-monotonic trends (e.g., Linear-MoF peaks at 192, iTransformer peaks at 336). This figure illustrates per-dataset behavior, not overall ranking.

*Figure 10.* Comparison of different normalization modules on the WEATHER dataset using DLinear. Similar to Fig. 8, this figure illustrates how input length interacts with normalization on a single dataset rather than providing an overall ranking.

### 4.5. MoF Generalization Analysis

**Rethinking Input Length** Most prior works fix input lengths at 96 or 336, but our experiments show input length strongly impacts performance. We evaluated the Weather and ETTh2 datasets across expanded horizons (48, 96, 192, 336, 720, 960 time steps) with a fixed prediction length of 192. Full results are in Appendix H. We compared Linear-MoF with other models (Fig. 9) and DLinear-based methods combined with various normalization modules (Fig. 10). Results indicate the optimal input length varies across models and datasets, likely due to the data's inherent frequency and structural bias. Notably, Linear-MoF achieves superior accuracy with fewer input steps and consistently outperforms other models at their respective optimal input lengths.

**Generalization of MoF Across Models** This section explores the integration of the MoF module with various backbone models, including Autoformer, Informer, and DLinear. Results indicate that incorporating the MoF module can reduce error rate by up to 58%, suggesting its potential to enhance the performance of diverse architectures. These findings highlight the adaptability of the MoF module as a promising approach for enhancing various models. Detailed results and analyses are available in Appendix J.

## 5. Conclusion

In this work, we demonstrate how fat-tailed distributions, prevalent in real-world temporal data, undermine forecast-

ing stability even at moderate excess kurtosis levels. The proposed *Morphing-Flow (MoF)* framework addresses this through two components: (1) a spline-based Flow layer that normalizes extremal values while preserving critical temporal patterns, and (2) a Morph module mitigating distributional drift via test-time adaptation. Experiments across 8 benchmarks in various settings validate MoF's effectiveness, as it surpasses competitors by 6.3% on average, achieving a 21.7% error reduction on fat-tailed datasets such as Exchange. Broader analysis confirms that MoF mitigates distributional skewness and temporal drift, promoting balanced gradient dynamics while maintaining computational efficiency and hyperparameter robustness. The plug-and-play design enables seamless integration into diverse architectures without structural changes, offering a lightweight deployment solution. These results suggest that adaptive handling of fat-tailed distributions is critical for reliable time series forecasting in non-stationary environments.

## Acknowledgements

This work was partly supported by the National Key Research and Development Program of China (No.2024YFB4711102). Meanwhile, this work was jointly funded by the Xiaomi Innovation Joint Fund of the Beijing Municipal Natural Science Foundation (No.L233006) and the project of Tsinghua University-Toyota Joint Research Center for AI Technology of Automated Vehicle (No. TTAD2025-03).

## Impact Statement

This paper presents work whose goal is to advance the field of Machine Learning. There are many potential societal consequences of our work, none which we feel must be specifically highlighted here.

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

# A. Related Work

**Fat-Tailed Features in Machine Learning.** Empirical evidence and theoretical analysis have revealed the presence of fat-tailed noise and leptokurtic features in modern machine learning systems (Simsekli et al., 2019; Nguyen et al., 2019). Such distributions challenge the bounded variance assumption in stochastic optimization, significantly impacting gradient stability and learning dynamics (Zhang et al., 2020; Wang et al., 2021). To address these challenges, methods like gradient clipping (Gorbunov et al., 2020; Yang et al., 2022) and adaptive normalization techniques for SGD (Hübler et al., 2024) have been proposed. Recent studies further demonstrate the prevalence of fat-tailed patterns across diverse domains, including economic indicators (Kiss & Österholm, 2020a), hydrological data (Wang et al., 2023), and epidemic transmission (Wong & Collins, 2020).

In financial indices, data with pronounced tails are commonly observed due to clustering volatility, a phenomenon linked to the heteroskedasticity of temporal sequences (Eom et al., 2019; Kiss & Österholm, 2020b). Models like GARCH (Bollerslev, 1986) are traditionally used to capture such behavior. As shown in Fig. 1, the heterogeneity of time series data results in consistently fat-tailed distributions across a wide range of domains. However, research on understanding and mitigating the effects of such characteristics in neural time series forecasting remains limited. MoF addresses this by reducing gradient skewness and kurtosis, particularly benefiting deeper backbones where outliers more severely disrupt training. Our results demonstrate that MoF yields substantially higher improvements with PatchTST than with shallow architectures like Linear, aligning with these domain-level insights.

**Non-Stationarity in Time Series Forecasting.** Time series forecasting typically faces time-variant phenomena and distribution shifts, often addressed through signal decomposition to separate stationary and non-stationary components (RB, 1990; Hamilton, 2020; Zeng et al., 2023). Ogasawara et al. (Ogasawara et al., 2010) proposed a local normalization approach, while RevIN (Kim et al., 2021) introduced reversible instance normalization to better handle distribution shifts at the sample level. Subsequent models have refined this direction, including Leddam's learnable convolution for trend extraction (Yu et al., 2024c), LiNo's recursive residual decomposition (Yu et al., 2024b), and FAN's frequency-based decomposition (Ye et al., 2024). While several models predict future statistical moments to better handle heteroskedasticity, they often underperform in the presence of fat-tailed noise, especially under limited batch-level statistics.

**Backbones and Generalization.** Time-slice-based transformers for time series forecasting struggle with weak temporal dependencies and noisy per-timestep encoding (Zhou et al., 2021; Kitaev et al., 2020). DLinear (Zeng et al., 2023) and iTransformer (Liu et al., 2023) initiated the shift toward modeling sequences holistically in temporal space. Patch-based architectures such as Crossformer (Zhang & Yan, 2023), TimesNet (Wu et al., 2022), and Mamba-based designs (Ahamed & Cheng, 2024; Zeng et al., 2024) further enhance modeling capacity but suffer from exposure to non-stationarity due to their reliance on localized features. Several studies have noted that existing forecasting benchmarks often fail to expose the generalization challenges of such models (Han et al., 2024; Ilbert et al., 2024).

Recent works like IN-Flow (Fan et al., 2025) tackle non-stationarity through entangled normalizing flows, but face scalability issues on high-dimensional datasets. FITS (Xu et al., 2023) offers parameter efficiency via hypernetwork-like reweighting, though it relies on expensive hyperparameter search. ElasTST (Zhang et al., 2024) proposes a backbone-agnostic design for multi-horizon generalization, which is orthogonal to our goal of improving normalization. Our experiments (Appendix K) show that MoF complements these design strategies and remains robust even under fixed hyperparameter settings across datasets.

# B. Synthetic Fat-tail Training Experiment

### B.1. Detailed Experiment Setup

**Dataset synthesis** The dataset consists of samples drawn from either a Gaussian or Cauchy distribution, where the mean grows at each step by a factor $k$. At step $t$, the mean of the distribution is defined as $\mu_t = k\mu_{t-1}$, with $\mu_0 = 1.0$. Each sample $x_t$ is generated as:

$$x_t = \mu_t + \epsilon_t, \quad \mu_t = k\mu_{t-1}, \quad t = 0, 1, \ldots, n-1$$

where $\epsilon_t$ is drawn from the respective distribution (Gaussian or Cauchy), and clipped to the range $[-6, 6]$. The dataset is structured such that the first $n$ samples are inputs, and the last one is the output:

$$y = x_n \quad \text{(output)}$$

**Simple Linear Model with Clipping** The model is a linear regression defined as:

$$\hat{y} = Wx + b$$

where $x \in \mathbb{R}^n$ is the input, and the output $\hat{y}$ is clipped within the range $[-20, 20]$ before being passed through the linear transformation. The parameters $W$ and $b$ are learned during training.

**Training Protocol** The model is trained using SGD with a learning rate $\eta = 0.01$ and Mean Squared Error (MSE) loss. A StepLR scheduler is used to reduce the learning rate by a factor of $0.1$ every 30 steps. The total number of iterations is calculated based on the batch size and dataset size, with each iteration consisting of a forward pass, loss computation, and backpropagation.

The loss function is:

$$\mathcal{L} = \frac{1}{B} \sum_{i=1}^{B} (\hat{y}_i - y_i)^2$$

where $B$ is the batch size.

**Hyperparameters** For both the Gaussian and Cauchy distributions, the following hyperparameters were used:

- **Mean growth factor** ($k$): 1.1
- **Batch size** ($B$): 128
- **Learning rate** ($\eta$): 0.05
- **Optimizer**: SGD
- **Loss function**: Mean Squared Error (MSE)
- **Learning rate scheduler**: StepLR with a step size of 30 steps and decay factor $\gamma = 0.1$
- **Total iterations**: 10,000

## B.2. More Results on Synthetic Datasets

Fig 11 are results with identical settings across multiple runs but different random initializations. These experiments reveal significant disparities in the convergence behavior of the models when trained on sequences from distributions with heavier tails.

## B.3. Stability Condition Analysis

A common way to express the stability condition of SGD with a (possibly non-convex) loss $\mathcal{L}$ is through:

$$\mathbb{E}\left[\|\theta_{t+1} - \theta^*\|^2\right] \leq \left(1 - 2\eta\,\lambda_{\min}(H)\right)^t \|\theta_0 - \theta^*\|^2 + \frac{\eta^2\,\sigma^2}{\lambda_{\min}(H)},$$

where $\theta^*$ is a stationary point, $H$ is the Hessian (or an equivalent notion of curvature), and $\sigma^2$ is an upper bound on the variance of the gradient noise. From an empirical standpoint:

- **Variance effect:** If $\sigma^2$ is large due to fat-tailed (though bounded) noise, the second term on the right-hand side grows, leading to poor convergence or large stationary error.

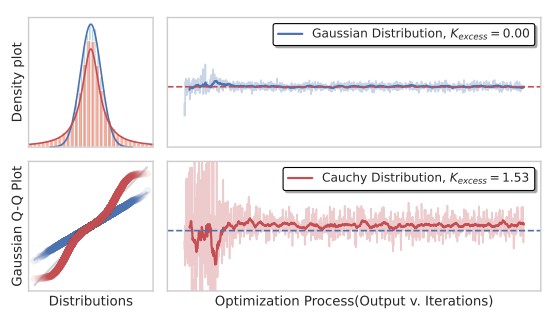

(a) Converges to an incorrect value above the target value.

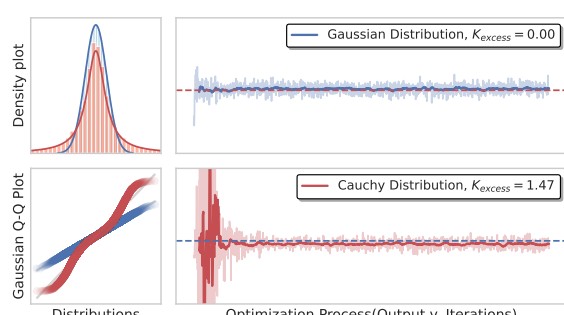

(b) Converges to a seemingly correct value, but still above the target value.

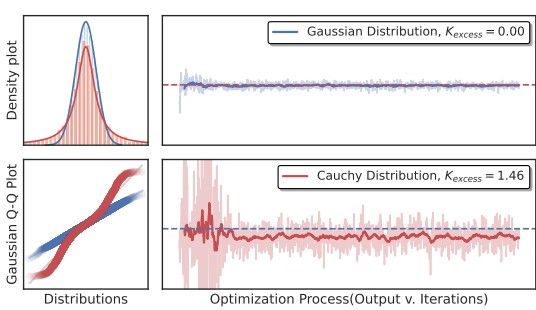

(c) Converges to an incorrect value below the target value.

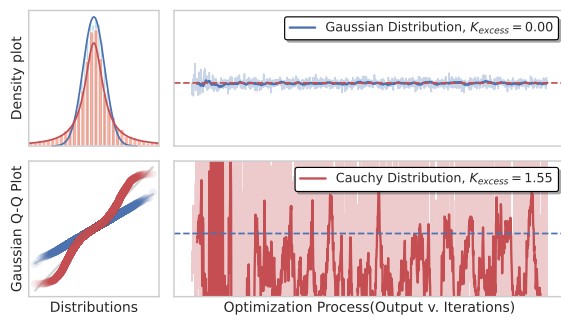

(d) Fails to converge entirely.

*Figure 11.* Training results on synthetic datasets with sequences from Gaussian and Cauchy distributions. Models trained on Gaussian sequences exhibit consistent convergence, while those trained on Cauchy sequences demonstrate significantly varied behaviors, ranging from incorrect convergence to complete failure. This highlights the sensitivity of models to fat-tailed distributions.

- **Starting point:** A poor initialization $\theta_0$ can exacerbate the challenge, especially when high-variance updates cause parameters to deviate before eventually converging to $\theta^*$. In real-world scenarios, we observed that replacing the DLinear model's "Average Initialization" with random initialization significantly affects model performance.

- **Other factors:** Learning rate $\eta$, local curvature $\lambda_{\min}(H)$, and problem structure (e.g., convex vs. non-convex) all empirically influence the extent to which fat-tailed noise hinders convergence.

## C. Dataset Overview

### C.1. Dataset Introduction

| Dataset | Granularity | Dataset Split | Prediction Length | Channels | Domain |
|---------|-------------|---------------|-------------------|----------|--------|
| ETTh1 | Hourly | (8545, 2881, 2881) | {96, 192, 336, 720} | 7 | Electricity |
| ETTh2 | Hourly | (8545, 2881, 2881) | {96, 192, 336, 720} | 7 | Electricity |
| ETTm1 | 15min | (34465, 11521, 11521) | {96, 192, 336, 720} | 7 | Electricity |
| ETTm2 | 15min | (34465, 11521, 11521) | {96, 192, 336, 720} | 7 | Electricity |
| Exchange | Daily | (5120, 665, 1422) | {96, 192, 336, 720} | 8 | Economy |
| Weather | 10min | (36792, 5271, 10540) | {96, 192, 336, 720} | 21 | Weather |
| Electricity | Hourly | (18317, 2633, 5261) | {96, 192, 336, 720} | 321 | Electricity |
| Traffic | Hourly | (12185, 1757, 3509) | {96, 192, 336, 720} | 862 | Transportation |
| NIL | Weekly | (580,193,193) | {24,36,48,60} | 7 | Healthcare |

- **ETT (Electricity Transformer Temperature)**[1]: This dataset includes four subsets: two hourly-level datasets (ETTh) and two 15-minute-level datasets (ETTm). Each subset consists of seven features related to the oil temperature and load of electricity transformers, recorded from July 2016 to July 2018.

- **Traffic**[2]: This dataset contains hourly road occupancy data collected from sensors on San Francisco freeways during 2015–2016.

- **Electricity**[3]: This dataset contains hourly electricity consumption data from 321 clients, recorded between 2012 and 2014.

- **Exchange-Rate**[4]: This dataset includes daily exchange rate data for 8 countries, spanning the years 1990 to 2016.

- **Weather**[5]: This dataset records 21 weather indicators, such as air temperature and humidity, at 10-minute intervals throughout 2020 in Germany.

- **ILI (Influenza-Like Illness)**[6]: The ILI dataset tracks the ratio of patients diagnosed with influenza-like illness to the total number of patients, based on weekly data from the Centers for Disease Control.

### C.2. Kurtosis Calculation

Kurtosis is a statistical measure used to describe the shape of a probability distribution, particularly its tails. It quantifies the extent to which the tails of a distribution differ from those of a normal distribution. The excess kurtosis $k_{\text{excess}}$ is calculated as:

$$k_{\text{excess}} = \frac{E[(X - \mu)^4]}{\sigma^4} - 3$$

where: - $X$ is a random variable, - $\mu$ is the mean of the distribution, - $\sigma$ is the standard deviation, and - $E[\cdot]$ denotes the expected value.

A kurtosis value of $k_{\text{excess}} = 0$ indicates a normal distribution, with heavier tails (leptokurtic) for positive values and lighter tails (platykurtic) for negative values.

Since each channel or dimension in a time series dataset represents a different feature with potentially different statistical properties, the distribution of each dimension may vary significantly. Therefore, kurtosis is computed separately for each

---

[1] https://arxiv.org/abs/2012.01655
[2] https://www.kaggle.com/datasets/jessemostipak/traffic
[3] https://archive.ics.uci.edu/ml/datasets/ElectricityLoadDiagrams20112014
[4] https://www.kaggle.com/datasets/loyolacoding/exchange-rate-data
[5] https://www.kaggle.com/datasets/muthuj7/weather-dataset
[6] https://www.cdc.gov/flu/weekly/index.htm

channel to accurately capture the distinct tail behaviors across features. This channel-wise analysis allows us to assess the presence of fat-tailed distributions or other characteristics specific to individual time-series features.

### C.3. Kurtosis Analysis

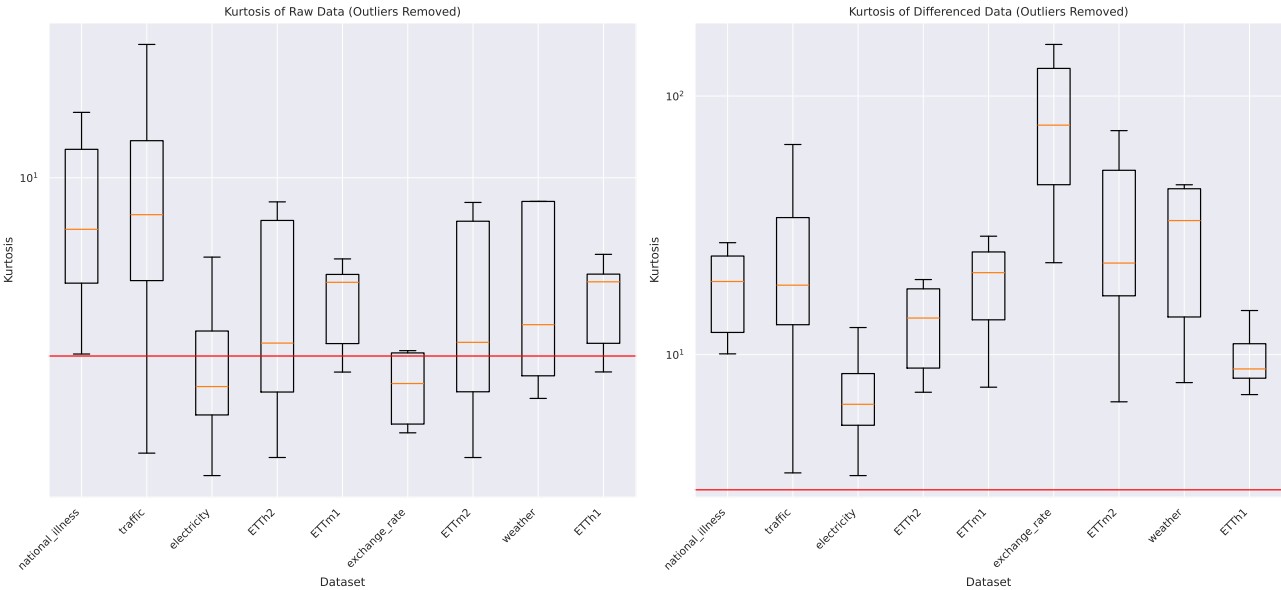

*Figure 12.* Kurtosis distribution across different channels. The red line represents a kurtosis of 3 ($K_{excess} = 0$), indicating a normal distribution.

We computed the kurtosis for each dimension across the datasets described earlier. The kurtosis distributions for **different channels** are illustrated in Fig. 12. Notably, the red line in the plot represents the theoretical kurtosis value of 3, which corresponds to a normal distribution (i.e., $k_{\text{excess}} = 0$). Deviations from this baseline indicate the degree of fat-tailedness present in the data.

The left plot presents the kurtosis distribution of the raw data, which are significant non-stationarity. Since the distribution appears dispersed across the feature space, the fat-tailed nature is not immediately evident. However, when we applied differentiation between adjacent time steps, the kurtosis values increased sharply, with most dimensions exhibiting a pronounced fat-tailed distribution. This observation underscores the pervasive fat-tailed nature of time-series data across different domains.

Although, the common practice of normalization using instance-wise z-scores tends to balance the feature space more evenly (in comparison to direct differencing of neighboring time steps). The kurtosis effect remains observable, reflecting the intrinsic properties of time-series data that persist across various datasets.

# D. Experiment Details

## D.1. Model Structure Details

**Stationary Method:** For preprocessing, we use a stationary method based on instance-wise z-score normalization. This process centers each instance by subtracting its mean and dividing by its standard deviation, ensuring that each input is normalized independently.

**Decomposition Method:** A standard moving average (MOA) kernel is used to decompose the trend and seasonal (residual) components of the data. The kernel size for the moving average is controlled by the `kernel_size` hyperparameter, which defaults to 25.

## D.2. Model Hyperparameters

This section provides a detailed overview of the hyperparameters used in various components of the model. Default values are specified for each parameter where applicable.

**MoF Hyperparameters:** For MoF modules the number of bins for the spline (`spline_num_bins`) is set to 24, the tail length of the spline (`spline_tail`) is set to 6.0, and the size of the b-matrix for Test Time Training (TT) is set to 92 for all the dataset. Additionally, the masking ratio $p$ for self-supervised Test Time Training (TTT) is set to 0.5.

Except for experiments involving long-range settings (e.g., prediction lengths of 336 and 720), where the dimension count is high and the distribution across channels is similar, due to computation power limitation, a same set of MoF parameters is shared across channels. In all other experiments, each channel utilizes dedicated MoF parameters.

**Patch Mamba Backbone Settings:** The Patch Mamba backbone is controlled by the following hyperparameters. The embedding dimension, denoted as `d_model`, is set to 64. Dropout is applied to the fully connected layers with a rate of 0.2 (`fc_dropout`), and to the attention heads with a rate of 0.0 (`head_dropout`). The padding method for patches is set to `end`, meaning that padding is applied at the sequence's end. Each patch has a length of 16 (`patch_len`), and the stride for patch extraction is set to 8 (`stride`).

**Linear Backbone Settings:** The output sequence is projected to the prediction length using a linear layer. The `Linear_Projection` layer projects the input sequence length (`seq_len`) to the prediction length (`pred_len`). The weights of the projection layer are initialized uniformly, with each element set to $\frac{1}{\texttt{seq\_len}}$, ensuring that all input elements are given equal importance at the start. This weight initialization method is consistent with previous approaches in the literature, such as those described in (Zeng et al., 2023). Although we found initialization can have an positive impact on model performance, as we have analyzed previously, this aspect is not a novel contribution of this work.

## D.3. Evaluation Protocol:

Following the protocol established by Autoformer (Wu et al., 2021), we use Mean Squared Error (MSE) and Mean Absolute Error (MAE) as the primary evaluation metrics. To account for variability in the results, each experiment is repeated five times. Since we cannot confirm the normality of the results, a non-parametric approach is employed to assess the statistical significance of performance differences. Specifically, we conduct Mann-Whitney U tests, which do not assume a normal distribution, to determine whether observed differences in performance are statistically significant .

For the main experiment, the historical horizon length is set to $T = 336$, with prediction lengths $F \in \{96, 192, 336, 720\}$, except for the National-Illness dataset, where $T = 60$ and $F \in \{24, 36, 48, 60\}$. For other baseline models, the default set of hyperparameters and evaluation protocol are used across different datasets, ensuring consistent performance comparison.

## D.4. Evaluation Protocol and Hyperparameter Settings

**Unified Hyperparameter Configuration.** Unlike prior works that tune hyperparameters per dataset, our experiments adopt a single, unified configuration across all datasets and methods—including our MoF variants and baseline models. This design choice eliminates confounding effects from dataset-specific tuning and ensures that performance differences more directly reflect model design. While this stricter setup may introduce small deviations from original reported numbers, it reduces the risk of overfitting to test sets and improves cross-dataset generalization.

**Comparison with Prior Works.** Previous models such as iTransformer and PatchTST employ different hyperparameter

choices for different datasets, as detailed in Table 4. These inconsistencies hinder fair comparison and obscure whether performance gains stem from model design or tuning. For example, the DLinear result reported in the iTransformer paper uses a short lookback length of 96, resulting in a shallow input (e.g., a 96×192 linear layer), which significantly underfits and degrades performance. In contrast, our re-implementations use a unified setup with lookback length fixed at 336 across all datasets and models, eliminating dataset-specific tuning. As shown in Table 4, this fairer and more challenging protocol improves generalization and yields stronger baselines, outperforming comparable reimplemented baselines by 1.69–4.35% in average MSE.

*Table 4.* Comparison of different time series forecasting methods (DLinear, iTransformer, PatchTST) reported in prior works versus our re-implementations under unified hyperparameter settings (rightmost group).

| Model | iT | PT | DL | PT | DL | iT | PT | DL |
|---|---|---|---|---|---|---|---|---|
| From paper: | | iTransformer | | | Patch TST | | Ours | |
| lookback length | 96 | 96 | 96 | 336 | 512 | | 336 | |
| Patch Len/Stride | - | 12/* | - | 16/8 to 24/2 | - | | 16/8 | |
| d_model | 128-512 | * | - | 16-128 | - | | 512 | |
| d_ff | 128-2048 | * | - | 128-256 | - | | 512 | |
| learning_rate | 1e-3 to 5e-5 | * | * | 2.5e-4 to 1e-4 | 1e-2 to 1e-4 | | 1E-04 | |
| batchsize | 16-32 | * | * | 24-128 | 8-32 | | 4 | |
| ETTh1 | 0.454 | 0.469 | 0.456 | 0.417 | 0.42 | 0.538 | 0.584 | 0.524 |
| ETTh2 | 0.383 | 0.387 | 0.559 | 0.331 | 0.43 | 0.261 | 0.263 | 0.228 |
| ETTm1 | 0.407 | 0.387 | 0.403 | 0.352 | 0.36 | 0.442 | 0.416 | 0.452 |
| ETTm2 | 0.288 | 0.281 | 0.350 | 0.258 | 0.27 | 0.182 | 0.169 | 0.170 |
| ECL | 0.178 | 0.205 | 0.212 | 0.162 | 0.17 | 0.165 | 0.164 | 0.167 |
| Exchange | 0.360 | 0.367 | 0.354 | - | - | 0.462 | 0.544 | 0.390 |
| Traffic | 0.428 | 0.481 | 0.625 | 0.396 | 0.43 | 0.451 | 0.416 | 0.435 |
| Weather | 0.258 | 0.259 | 0.265 | 0.230 | 0.25 | 0.241 | 0.234 | 0.245 |
| **Average** | 0.345 | 0.353 | 0.403 | 0.307 | 0.33 | 0.343 (2.72%) | 0.349 (4.35%) | 0.326 (1.69%) |

*Abbreviations: iT=iTransformer, PT=PatchTST, DL=DLinear, *=not available

**Baseline Alignment.** To validate our experimental setup, we reproduce results for low-capacity models like DLinear and compare them with original and recent benchmark papers (Table 5). Our results align well across all datasets.

*Table 5.* DLinear MSE comparison across papers.

| Dataset | Ours | DLinear (AAAI'23) | LIFT (ICLR'24) | SAN (NeurIPS'23) |
|---|---|---|---|---|
| Weather | 0.245 | 0.246 | 0.246 | 0.245 |
| Electricity | 0.167 | 0.166 | 0.166 | 0.166 |
| Traffic | 0.435 | 0.434 | 0.434 | 0.435 |

**Stronger Baselines under Fair Settings.** Even under this more stringent unified protocol, our re-implementations of strong baselines achieve better results than originally reported—e.g., our iTransformer implementation improves over the original by 1.69% to 19.1% in average MSE. This highlights the influence of consistent experimental conditions.

*Table 6.* Average MSE performance under unified settings.

| Model | Ours (MSE) | iTransformer Paper | PatchTST Paper |
|---|---|---|---|
| iTransformer | 0.343 (**−2.72%**) | 0.345 | – |
| PatchTST | 0.349 (**−4.35%**) | 0.353 | 0.307 |
| DLinear | 0.326 (**−19.1%**) | 0.403 | 0.330 |

**MoF Gains Remain Robust.** Despite the stronger baselines, our proposed MoF module—especially when used with the

Mamba backbone—consistently improves over both iTransformer and PatchTST by 14.9% and 16.9% in average MSE, respectively.

# E. Detailed Comprehensive Experiment Results

### E.1. MAE and MSE performances

We compare MoF with Mamba and Linear backbones against 10 state-of-the-art models or widely acknowledged methods. Each experiment is repeated five times, and the best average results are shown in black, while the second-best results are highlighted in blue and underlined.

Due to space limitations, the experiment results are presented across two tables.

*Table 7.* Model Performance Table

| Model Dataset | | MoF-Mamba MAE | MSE | MoF-Linear MAE | MSE | GLinear MAE | MSE | LiNo MAE | MSE | TimeMachine MAE | MSE | CATS MAE | MSE | iTransformer MAE | MSE |
|---|---|---|---|---|---|---|---|---|---|---|---|---|---|---|---|
| ETTh1 | 96 | 0.441 | **0.406** | **0.439** | 0.416 | 0.446 | 0.419 | 0.446 | 0.411 | 0.445 | 0.417 | 0.584 | 0.664 | 0.465 | 0.439 |
| | 192 | 0.480 | **0.458** | **0.479** | 0.467 | 0.485 | 0.472 | 0.486 | 0.465 | 0.485 | 0.472 | 0.618 | 0.722 | 0.505 | 0.491 |
| | 336 | 0.522 | **0.513** | **0.514** | 0.519 | 0.521 | 0.526 | 0.531 | 0.531 | 0.527 | 0.534 | 0.669 | 0.834 | 0.540 | 0.547 |
| | 720 | 0.595 | **0.627** | 0.603 | 0.658 | 0.612 | 0.673 | 0.638 | 0.707 | 0.625 | 0.695 | 0.772 | 1.066 | 0.618 | 0.675 |
| | Avg. | 0.510 | **0.501** | **0.509** | 0.515 | 0.516 | 0.522 | 0.525 | 0.528 | 0.520 | 0.529 | 0.661 | 0.822 | 0.532 | 0.538 |
| ETTh2 | 96 | 0.286 | 0.173 | **0.279** | **0.166** | 0.288 | 0.175 | 0.290 | 0.175 | 0.290 | 0.177 | 0.315 | 0.198 | 0.297 | 0.189 |
| | 192 | 0.314 | 0.207 | **0.310** | **0.204** | 0.320 | 0.213 | 0.324 | 0.217 | 0.322 | 0.216 | 0.344 | 0.235 | 0.333 | 0.233 |
| | 336 | 0.338 | **0.236** | 0.337 | 0.238 | 0.350 | 0.250 | 0.359 | 0.262 | 0.348 | 0.249 | 0.372 | 0.273 | 0.359 | 0.270 |
| | 720 | 0.394 | **0.281** | **0.371** | 0.282 | 0.400 | 0.322 | 0.414 | 0.340 | 0.399 | 0.324 | 0.420 | 0.347 | 0.410 | 0.353 |
| | Avg. | 0.333 | 0.224 | **0.324** | **0.222** | 0.340 | 0.240 | 0.347 | 0.248 | 0.340 | 0.241 | 0.363 | 0.263 | 0.350 | 0.261 |
| ETTm1 | 96 | 0.376 | 0.324 | 0.379 | 0.328 | **0.368** | **0.315** | 0.380 | 0.327 | 0.370 | 0.321 | 0.393 | 0.342 | 0.398 | 0.361 |
| | 192 | 0.402 | 0.368 | 0.412 | 0.377 | **0.401** | 0.368 | 0.413 | 0.378 | 0.407 | 0.376 | 0.425 | 0.392 | 0.432 | 0.416 |
| | 336 | **0.433** | **0.407** | 0.441 | 0.419 | 0.433 | 0.415 | 0.448 | 0.428 | 0.438 | 0.420 | 0.450 | 0.430 | 0.462 | 0.462 |
| | 720 | 0.484 | **0.468** | 0.486 | 0.479 | **0.479** | 0.481 | 0.494 | 0.495 | 0.489 | 0.495 | 0.495 | 0.493 | 0.507 | 0.529 |
| | Avg. | 0.424 | **0.392** | 0.429 | 0.401 | **0.420** | 0.395 | 0.434 | 0.407 | 0.426 | 0.403 | 0.441 | 0.414 | 0.450 | 0.442 |
| ETTm2 | 96 | **0.226** | **0.113** | 0.228 | 0.118 | 0.229 | 0.115 | 0.228 | 0.114 | 0.226 | 0.114 | 0.235 | 0.118 | 0.240 | 0.127 |
| | 192 | **0.256** | **0.146** | 0.263 | 0.165 | 0.259 | 0.148 | 0.260 | 0.147 | 0.258 | 0.148 | 0.266 | 0.149 | 0.271 | 0.161 |
| | 336 | **0.282** | **0.175** | 0.292 | 0.200 | 0.286 | 0.179 | 0.287 | 0.180 | 0.284 | 0.180 | 0.298 | 0.186 | 0.302 | 0.200 |
| | 720 | 0.326 | 0.227 | 0.329 | 0.248 | **0.319** | **0.220** | 0.322 | 0.224 | 0.319 | 0.220 | 0.335 | 0.233 | 0.335 | 0.242 |
| | Avg. | 0.273 | **0.165** | 0.278 | 0.183 | 0.273 | 0.166 | 0.274 | 0.166 | **0.272** | 0.165 | 0.283 | 0.172 | 0.287 | 0.182 |
| Electricity | 96 | 0.228 | 0.132 | 0.233 | 0.137 | 0.232 | 0.136 | 0.243 | 0.143 | 0.229 | 0.134 | 0.280 | 0.184 | 0.228 | 0.132 |
| | 192 | 0.242 | 0.149 | 0.246 | 0.151 | 0.247 | 0.153 | 0.259 | 0.161 | 0.244 | 0.152 | 0.297 | 0.209 | 0.249 | 0.153 |
| | 336 | **0.259** | **0.165** | 0.262 | 0.167 | 0.266 | 0.171 | 0.276 | 0.178 | 0.262 | 0.169 | 0.312 | 0.225 | 0.263 | 0.167 |
| | 720 | 0.294 | 0.205 | **0.293** | **0.201** | 0.298 | 0.209 | 0.308 | 0.217 | 0.295 | 0.207 | 0.342 | 0.264 | 0.301 | 0.209 |
| | Avg. | 0.256 | **0.163** | 0.259 | 0.164 | 0.261 | 0.167 | 0.272 | 0.175 | 0.258 | 0.166 | 0.308 | 0.221 | 0.260 | 0.165 |
| Exchange | 96 | 0.214 | 0.088 | 0.211 | 0.092 | 0.214 | 0.092 | 0.214 | 0.092 | 0.216 | 0.094 | 0.212 | 0.093 | 0.242 | 0.113 |
| | 192 | 0.315 | 0.186 | 0.310 | 0.194 | 0.316 | 0.196 | 0.317 | 0.198 | 0.325 | 0.208 | 0.309 | 0.192 | 0.356 | 0.240 |
| | 336 | **0.426** | **0.325** | 0.434 | 0.363 | 0.436 | 0.362 | 0.446 | 0.375 | 0.447 | 0.383 | 0.431 | 0.354 | 0.470 | 0.409 |
| | 720 | **0.621** | **0.621** | 0.729 | 0.958 | 0.717 | 0.924 | 0.721 | 0.949 | 0.734 | 1.042 | 0.737 | 0.948 | 0.773 | 1.084 |
| | Avg. | **0.394** | **0.305** | 0.421 | 0.402 | 0.421 | 0.394 | 0.424 | 0.403 | 0.430 | 0.432 | 0.422 | 0.397 | 0.460 | 0.462 |
| Traffic | 96 | 0.271 | 0.391 | 0.279 | 0.407 | 0.284 | 0.400 | 0.312 | 0.425 | 0.273 | **0.389** | 0.457 | 0.691 | 0.286 | 0.419 |
| | 192 | 0.277 | **0.407** | 0.284 | 0.420 | 0.293 | 0.418 | 0.326 | 0.450 | 0.282 | 0.408 | 0.485 | 0.750 | 0.293 | 0.439 |
| | 336 | 0.284 | 0.420 | 0.291 | 0.432 | 0.301 | 0.432 | 0.332 | 0.464 | 0.291 | 0.422 | 0.661 | 1.072 | 0.302 | 0.457 |
| | 720 | 0.300 | **0.447** | 0.313 | 0.463 | 0.318 | 0.461 | 0.345 | 0.491 | 0.309 | 0.451 | 0.735 | 1.235 | 0.322 | 0.490 |
| | Avg. | 0.283 | 0.416 | 0.292 | 0.430 | 0.299 | 0.428 | 0.329 | 0.458 | 0.289 | 0.417 | 0.585 | 0.937 | 0.301 | 0.451 |
| Weather | 96 | **0.196** | **0.145** | 0.202 | 0.149 | 0.200 | 0.148 | 0.199 | 0.155 | 0.199 | 0.148 | 0.201 | 0.147 | 0.214 | 0.163 |
| | 192 | **0.237** | **0.186** | 0.245 | 0.193 | 0.241 | 0.191 | 0.246 | 0.207 | 0.241 | 0.193 | 0.242 | 0.191 | 0.255 | 0.209 |
| | 336 | **0.278** | **0.238** | 0.282 | 0.239 | 0.281 | 0.243 | 0.287 | 0.258 | 0.281 | 0.243 | 0.282 | 0.243 | 0.292 | 0.260 |
| | 720 | **0.332** | 0.317 | 0.333 | **0.308** | 0.334 | 0.321 | 0.340 | 0.334 | 0.335 | 0.326 | 0.334 | 0.318 | 0.342 | 0.333 |
| | Avg. | **0.261** | **0.221** | 0.266 | 0.222 | 0.264 | 0.226 | 0.268 | 0.238 | 0.264 | 0.227 | 0.265 | 0.225 | 0.276 | 0.241 |

*Table 8.* Model Performance Table (cont.)

| | Model | MoF-Mamba | | MoF-Linear | | PatchTST | | DLinear | | Autoformer | | Informer | | Transformer | |
|---|---|---|---|---|---|---|---|---|---|---|---|---|---|---|---|
| | Dataset | MAE | MSE | MAE | MSE | MAE | MSE | MAE | MSE | MAE | MSE | MAE | MSE | MAE | MSE |
| ETTh1 | 96 | 0.441 | **0.406** | **0.439** | 0.416 | 0.476 | 0.459 | 0.444 | 0.436 | 0.596 | 0.649 | 0.834 | 1.226 | 0.625 | 0.699 |
| | 192 | 0.480 | **0.458** | **0.479** | 0.467 | 0.524 | 0.529 | 0.483 | 0.486 | 0.635 | 0.735 | 0.867 | 1.328 | 0.671 | 0.815 |
| | 336 | 0.522 | **0.513** | **0.514** | 0.519 | 0.565 | 0.596 | 0.518 | 0.531 | 0.690 | 0.843 | 0.786 | 1.121 | 0.686 | 0.829 |
| | 720 | 0.595 | **0.627** | 0.603 | 0.658 | 0.653 | 0.751 | **0.594** | 0.645 | 0.705 | 0.857 | 0.864 | 1.295 | 0.739 | 0.970 |
| | Avg. | 0.510 | **0.501** | **0.509** | 0.515 | 0.554 | 0.584 | 0.510 | 0.524 | 0.656 | 0.771 | 0.838 | 1.243 | 0.680 | 0.828 |
| ETTh2 | 96 | 0.286 | 0.173 | **0.279** | **0.166** | 0.302 | 0.189 | 0.285 | 0.176 | 0.367 | 0.248 | 0.440 | 0.362 | 0.421 | 0.323 |
| | 192 | 0.314 | 0.207 | **0.310** | **0.204** | 0.340 | 0.235 | 0.316 | 0.213 | 0.407 | 0.302 | 0.484 | 0.439 | 0.446 | 0.381 |
| | 336 | 0.338 | **0.236** | 0.337 | 0.238 | 0.372 | 0.277 | 0.342 | 0.236 | 0.442 | 0.349 | 0.478 | 0.434 | 0.474 | 0.399 |
| | 720 | 0.394 | **0.281** | 0.371 | 0.282 | 0.424 | 0.349 | 0.386 | 0.287 | 0.566 | 0.598 | 0.578 | 0.599 | 0.607 | 0.620 |
| | Avg. | 0.333 | 0.224 | **0.324** | **0.222** | 0.359 | 0.263 | 0.332 | 0.228 | 0.446 | 0.374 | 0.495 | 0.458 | 0.487 | 0.431 |
| ETTm1 | 96 | 0.376 | 0.324 | 0.379 | 0.328 | 0.385 | 0.336 | 0.386 | 0.355 | 0.559 | 0.617 | 0.558 | 0.621 | 0.508 | 0.523 |
| | 192 | 0.402 | **0.368** | 0.412 | 0.377 | 0.425 | 0.394 | 0.420 | 0.421 | 0.556 | 0.583 | 0.609 | 0.703 | 0.550 | 0.585 |
| | 336 | **0.433** | **0.407** | 0.441 | 0.419 | 0.457 | 0.434 | 0.456 | 0.485 | 0.569 | 0.609 | 0.658 | 0.783 | 0.614 | 0.698 |
| | 720 | 0.484 | **0.468** | 0.486 | 0.479 | 0.502 | 0.502 | 0.502 | 0.547 | 0.606 | 0.668 | 0.706 | 0.872 | 0.668 | 0.813 |
| | Avg. | 0.424 | **0.392** | 0.429 | 0.401 | 0.442 | 0.416 | 0.441 | 0.452 | 0.573 | 0.619 | 0.633 | 0.745 | 0.585 | 0.655 |
| ETTm2 | 96 | **0.226** | **0.113** | 0.228 | 0.118 | 0.230 | 0.117 | 0.234 | 0.120 | 0.283 | 0.163 | 0.308 | 0.208 | 0.288 | 0.165 |
| | 192 | **0.256** | **0.146** | 0.263 | 0.165 | 0.259 | 0.147 | 0.263 | 0.148 | 0.298 | 0.181 | 0.367 | 0.292 | 0.343 | 0.242 |
| | 336 | **0.282** | **0.175** | 0.292 | 0.200 | 0.290 | 0.183 | 0.292 | 0.181 | 0.315 | 0.205 | 0.404 | 0.350 | 0.403 | 0.308 |
| | 720 | 0.326 | 0.227 | 0.329 | 0.248 | 0.328 | 0.230 | 0.333 | 0.232 | 0.347 | 0.247 | 0.550 | 0.672 | 0.481 | 0.455 |
| | Avg. | 0.273 | **0.165** | 0.278 | 0.183 | 0.277 | 0.169 | 0.280 | 0.170 | 0.311 | 0.199 | 0.407 | 0.381 | 0.379 | 0.293 |
| Electricity | 96 | 0.228 | 0.132 | 0.233 | 0.137 | **0.226** | **0.132** | 0.238 | 0.140 | 0.316 | 0.203 | 0.418 | 0.344 | 0.373 | 0.283 |
| | 192 | 0.242 | 0.149 | 0.246 | 0.151 | **0.242** | **0.149** | 0.251 | 0.154 | 0.329 | 0.213 | 0.433 | 0.364 | 0.390 | 0.304 |
| | 336 | **0.259** | **0.165** | 0.262 | 0.167 | 0.260 | 0.166 | 0.268 | 0.169 | 0.334 | 0.219 | 0.433 | 0.367 | 0.379 | 0.292 |
| | 720 | 0.294 | 0.205 | **0.293** | **0.201** | 0.295 | 0.207 | 0.301 | 0.204 | 0.352 | 0.247 | 0.415 | 0.342 | 0.374 | 0.287 |
| | Avg. | 0.256 | **0.163** | 0.259 | 0.164 | **0.256** | 0.164 | 0.265 | 0.167 | 0.333 | 0.221 | 0.425 | 0.354 | 0.379 | 0.291 |
| Exchange | 96 | 0.214 | 0.088 | 0.211 | 0.092 | 0.260 | 0.198 | **0.209** | **0.084** | 0.629 | 0.660 | 0.898 | 1.159 | 0.835 | 1.091 |
| | 192 | 0.315 | 0.186 | 0.310 | 0.194 | 0.372 | 0.321 | **0.307** | **0.171** | 0.900 | 1.274 | 0.973 | 1.356 | 0.912 | 1.268 |
| | 336 | **0.426** | **0.325** | 0.434 | 0.363 | 0.512 | 0.581 | 0.435 | 0.332 | 0.916 | 1.313 | 1.049 | 1.635 | 0.994 | 1.505 |
| | 720 | **0.621** | **0.621** | 0.729 | 0.958 | 0.749 | 1.074 | 0.743 | 0.971 | 0.947 | 1.547 | 1.315 | 2.553 | 0.812 | 1.030 |
| | Avg. | **0.394** | **0.305** | 0.421 | 0.402 | 0.473 | 0.544 | 0.423 | 0.390 | 0.848 | 1.198 | 1.059 | 1.676 | 0.888 | 1.224 |
| Traffic | 96 | 0.271 | 0.391 | 0.279 | 0.407 | **0.268** | 0.390 | 0.285 | 0.411 | 0.384 | 0.637 | 0.445 | 0.793 | 0.382 | 0.702 |
| | 192 | 0.277 | **0.407** | 0.284 | 0.420 | **0.274** | 0.407 | 0.290 | 0.424 | 0.397 | 0.660 | 0.416 | 0.744 | 0.382 | 0.696 |
| | 336 | 0.284 | 0.420 | 0.291 | 0.432 | **0.283** | **0.419** | 0.298 | 0.437 | 0.377 | 0.628 | 0.488 | 0.894 | 0.369 | 0.686 |
| | 720 | 0.300 | **0.447** | 0.313 | 0.463 | **0.299** | 0.448 | 0.317 | 0.467 | 0.406 | 0.668 | 0.450 | 0.815 | 0.364 | 0.681 |
| | Avg. | 0.283 | 0.416 | 0.292 | 0.430 | **0.281** | **0.416** | 0.297 | 0.435 | 0.391 | 0.648 | 0.450 | 0.811 | 0.374 | 0.691 |
| Weather | 96 | **0.196** | **0.145** | 0.202 | 0.149 | 0.200 | 0.151 | 0.235 | 0.174 | 0.365 | 0.290 | 0.357 | 0.290 | 0.344 | 0.285 |
| | 192 | **0.237** | **0.186** | 0.245 | 0.193 | 0.245 | 0.198 | 0.276 | 0.217 | 0.399 | 0.342 | 0.400 | 0.356 | 0.460 | 0.445 |
| | 336 | **0.278** | **0.238** | 0.282 | 0.239 | 0.290 | 0.257 | 0.316 | 0.263 | 0.422 | 0.376 | 0.563 | 0.621 | 0.545 | 0.598 |
| | 720 | **0.332** | 0.317 | 0.333 | **0.308** | 0.341 | 0.330 | 0.363 | 0.325 | 0.445 | 0.438 | 0.666 | 0.850 | 0.604 | 0.694 |
| | Avg. | **0.261** | **0.221** | 0.266 | 0.222 | 0.269 | 0.234 | 0.297 | 0.245 | 0.408 | 0.361 | 0.496 | 0.529 | 0.488 | 0.505 |

### E.2. Mann–Whitney U Test

Since each experiment is repeated five times and we cannot assume the normality of the result distributions, we conduct a one-tailed Mann–Whitney U test for each model against the best-performing model in that configuration across 10 models. The p-values of the tests are presented in Table 9, where p-values greater than or equal to 0.95 are highlighted in **bold**, and p-values greater than 0.05 are highlighted in blue.

Interestingly, when tested on these models, the distribution of p-values across different datasets reveals that the ETTm datasets has the poorest discriminatory ability, followed by the Exchange dataset, while the ETTh series, Traffic, and Weather datasets demonstrate better discriminatory ability.

Due to space limitations, the experiment results are presented across two tables.

*Table 9.* Mann–Whitney U test

| Model / Dataset | MoF-Mamba MAE | MSE | MoF-Linear MAE | MSE | GLinear MAE | MSE | LiNo MAE | MSE | TimeMachine MAE | MSE | CATS MAE | MSE | iTransformer MAE | MSE |
|---|---|---|---|---|---|---|---|---|---|---|---|---|---|---|
| ETTh1 96 | 0.238 | **1.000** | **1.000** | 0.005 | 0.003 | 0.001 | 0.005 | 0.008 | 0.003 | 0.001 | 0.005 | 0.001 | 0.008 | 0.002 |
| ETTh1 192 | 0.393 | **1.000** | **1.000** | 0.018 | 0.029 | 0.008 | 0.029 | 0.016 | 0.012 | 0.002 | 0.018 | 0.004 | 0.018 | 0.004 |
| ETTh1 336 | 0.200 | **1.000** | **1.000** | 0.114 | 0.029 | 0.014 | 0.018 | 0.008 | 0.012 | 0.005 | 0.018 | 0.008 | 0.018 | 0.008 |
| ETTh1 720 | 0.393 | **1.000** | 0.018 | 0.050 | 0.008 | 0.029 | 0.004 | 0.018 | 0.002 | 0.012 | 0.004 | 0.018 | 0.004 | 0.018 |
| ETTh2 96 | 0.029 | 0.029 | **1.000** | **1.000** | 0.003 | 0.003 | 0.003 | 0.003 | 0.003 | 0.003 | 0.005 | 0.005 | 0.005 | 0.005 |
| ETTh2 192 | 0.029 | 0.029 | **1.000** | **1.000** | 0.029 | 0.029 | 0.029 | 0.029 | 0.012 | 0.012 | 0.018 | 0.018 | 0.012 | 0.012 |
| ETTh2 336 | 0.350 | **1.000** | **1.000** | 0.200 | 0.029 | 0.029 | 0.018 | 0.018 | 0.012 | 0.012 | 0.018 | 0.018 | 0.012 | 0.012 |
| ETTh2 720 | 0.050 | **1.000** | **1.000** | 0.500 | 0.029 | 0.029 | 0.018 | 0.018 | 0.012 | 0.012 | 0.018 | 0.018 | 0.018 | 0.018 |
| ETTm1 96 | 0.002 | 0.002 | 0.005 | 0.005 | **1.000** | **1.000** | 0.001 | 0.001 | 0.011 | 0.001 | 0.002 | 0.002 | 0.001 | 0.001 |
| ETTm1 192 | 0.171 | **1.000** | 0.029 | 0.029 | **1.000** | 0.243 | 0.014 | 0.014 | 0.005 | 0.005 | 0.008 | 0.008 | 0.005 | 0.005 |
| ETTm1 336 | **1.000** | **1.000** | 0.050 | 0.050 | 0.314 | 0.029 | 0.018 | 0.018 | 0.012 | 0.012 | 0.018 | 0.018 | 0.012 | 0.012 |
| ETTm1 720 | 0.114 | **1.000** | 0.114 | 0.050 | **1.000** | 0.029 | 0.008 | 0.018 | 0.019 | 0.012 | 0.014 | 0.029 | 0.005 | 0.012 |
| ETTm2 96 | **1.000** | **1.000** | 0.243 | 0.029 | 0.143 | 0.143 | 0.129 | 0.305 | 0.842 | 0.324 | 0.008 | 0.008 | 0.005 | 0.005 |
| ETTm2 192 | **1.000** | **1.000** | 0.018 | 0.018 | 0.032 | 0.056 | 0.032 | 0.365 | 0.214 | 0.089 | 0.004 | 0.016 | 0.002 | 0.002 |
| ETTm2 336 | **1.000** | **1.000** | 0.100 | 0.100 | 0.067 | 0.067 | 0.048 | 0.048 | 0.143 | 0.036 | 0.048 | 0.048 | 0.036 | 0.036 |
| ETTm2 720 | 0.067 | 0.600 | 0.029 | 0.029 | **1.000** | **1.000** | 0.032 | 0.056 | 0.381 | 0.381 | 0.014 | 0.014 | 0.005 | 0.005 |
| Electricity 96 | 0.083 | 0.083 | 0.012 | 0.012 | 0.005 | 0.005 | 0.005 | 0.005 | 0.001 | 0.001 | 0.002 | 0.002 | 0.004 | 0.242 |
| Electricity 192 | 0.571 | 0.571 | 0.012 | 0.012 | 0.005 | 0.005 | 0.036 | 0.036 | 0.001 | 0.001 | 0.002 | 0.002 | 0.001 | 0.001 |
| Electricity 336 | **1.000** | **1.000** | 0.100 | 0.100 | 0.067 | 0.067 | 0.048 | 0.048 | 0.036 | 0.036 | 0.048 | 0.048 | 0.036 | 0.036 |
| Electricity 720 | 0.600 | 0.100 | **1.000** | **1.000** | 0.050 | 0.050 | 0.018 | 0.018 | 0.018 | 0.018 | 0.029 | 0.029 | 0.012 | 0.012 |
| Exchange 96 | 0.038 | 0.020 | 0.152 | 0.001 | 0.011 | 0.000 | 0.021 | 0.000 | 0.005 | 0.000 | 0.155 | 0.001 | 0.000 | 0.000 |
| Exchange 192 | 0.030 | 0.011 | 0.261 | 0.002 | 0.011 | 0.000 | 0.030 | 0.000 | 0.002 | 0.000 | 0.168 | 0.000 | 0.000 | 0.000 |
| Exchange 336 | **1.000** | **1.000** | 0.125 | 0.018 | 0.095 | 0.008 | 0.028 | 0.004 | 0.015 | 0.002 | 0.274 | 0.004 | 0.002 | 0.002 |
| Exchange 720 | **1.000** | **1.000** | 0.029 | 0.029 | 0.014 | 0.014 | 0.008 | 0.008 | 0.005 | 0.005 | 0.014 | 0.014 | 0.005 | 0.005 |
| Traffic 96 | 0.001 | 0.002 | 0.003 | 0.005 | 0.001 | 0.002 | 0.001 | 0.002 | 0.001 | **1.000** | 0.001 | 0.002 | 0.000 | 0.000 |
| Traffic 192 | 0.005 | **1.000** | 0.012 | 0.029 | 0.005 | 0.014 | 0.012 | 0.029 | 0.001 | 0.129 | 0.002 | 0.008 | 0.001 | 0.003 |
| Traffic 336 | 0.048 | 0.286 | 0.018 | 0.018 | 0.008 | 0.008 | 0.006 | 0.006 | 0.002 | 0.002 | 0.004 | 0.004 | 0.008 | 0.008 |
| Traffic 720 | 0.267 | **1.000** | 0.029 | 0.100 | 0.014 | 0.067 | 0.008 | 0.048 | 0.008 | 0.048 | 0.014 | 0.067 | 0.029 | 0.100 |
| Weather 96 | **1.000** | **1.000** | 0.029 | 0.029 | 0.008 | 0.008 | 0.012 | 0.012 | 0.008 | 0.008 | 0.012 | 0.012 | 0.018 | 0.018 |
| Weather 192 | **1.000** | **1.000** | 0.050 | 0.050 | 0.029 | 0.029 | 0.029 | 0.029 | 0.012 | 0.012 | 0.018 | 0.018 | 0.018 | 0.018 |
| Weather 336 | **1.000** | **1.000** | 0.100 | 0.600 | 0.067 | 0.067 | 0.048 | 0.048 | 0.036 | 0.036 | 0.048 | 0.048 | 0.036 | 0.036 |
| Weather 720 | **1.000** | 0.100 | 0.400 | **1.000** | 0.600 | 0.029 | 0.048 | 0.018 | 0.071 | 0.012 | 0.067 | 0.029 | 0.036 | 0.012 |

Table 10. Mann–Whitney U test (cont.)

| Model Dataset | | MoF-Mamba MAE | MSE | MoF-Linear MAE | MSE | PatchTST MAE | MSE | DLinear MAE | MSE | Autoformer MAE | MSE | Informer MAE | MSE | Transformer MAE | MSE |
|---|---|---|---|---|---|---|---|---|---|---|---|---|---|---|---|
| ETTh1 | 96 | 0.238 | **1.000** | **1.000** | 0.005 | 0.001 | 0.000 | 0.008 | 0.002 | 0.008 | 0.002 | 0.005 | 0.001 | 0.005 | 0.001 |
| | 192 | 0.393 | **1.000** | **1.000** | 0.018 | 0.002 | 0.000 | 0.018 | 0.004 | 0.018 | 0.004 | 0.012 | 0.002 | 0.012 | 0.002 |
| | 336 | 0.200 | **1.000** | **1.000** | 0.114 | 0.002 | 0.001 | 0.036 | 0.008 | 0.018 | 0.008 | 0.012 | 0.005 | 0.012 | 0.005 |
| | 720 | 0.393 | **1.000** | 0.018 | 0.050 | 0.000 | 0.002 | **1.000** | 0.018 | 0.004 | 0.018 | 0.002 | 0.012 | 0.002 | 0.012 |
| ETTh2 | 96 | 0.029 | 0.029 | **1.000** | **1.000** | 0.001 | 0.001 | 0.008 | 0.008 | 0.014 | 0.014 | 0.008 | 0.008 | 0.008 | 0.008 |
| | 192 | 0.029 | 0.029 | **1.000** | **1.000** | 0.003 | 0.003 | 0.018 | 0.018 | 0.018 | 0.018 | 0.018 | 0.018 | 0.012 | 0.012 |
| | 336 | 0.350 | **1.000** | **1.000** | 0.200 | 0.003 | 0.003 | 0.018 | 0.286 | 0.018 | 0.018 | 0.018 | 0.018 | 0.012 | 0.012 |
| | 720 | 0.050 | **1.000** | **1.000** | 0.500 | 0.003 | 0.003 | 0.018 | 0.018 | 0.018 | 0.018 | 0.018 | 0.018 | 0.012 | 0.012 |
| ETTm1 | 96 | 0.002 | 0.002 | 0.005 | 0.005 | 0.000 | 0.000 | 0.002 | 0.002 | 0.002 | 0.002 | 0.001 | 0.001 | 0.001 | 0.001 |
| | 192 | 0.171 | **1.000** | 0.029 | 0.029 | 0.002 | 0.002 | 0.008 | 0.008 | 0.008 | 0.008 | 0.005 | 0.005 | 0.005 | 0.005 |
| | 336 | **1.000** | **1.000** | 0.050 | 0.050 | 0.006 | 0.006 | 0.018 | 0.018 | 0.029 | 0.029 | 0.012 | 0.012 | 0.018 | 0.018 |
| | 720 | 0.114 | **1.000** | 0.114 | 0.050 | 0.002 | 0.006 | 0.008 | 0.018 | 0.014 | 0.029 | 0.005 | 0.012 | 0.008 | 0.018 |
| ETTm2 | 96 | **1.000** | **1.000** | 0.243 | 0.029 | 0.129 | 0.019 | 0.008 | 0.008 | 0.014 | 0.014 | 0.008 | 0.008 | 0.008 | 0.008 |
| | 192 | **1.000** | **1.000** | 0.018 | 0.018 | 0.026 | 0.214 | 0.004 | 0.016 | 0.008 | 0.008 | 0.004 | 0.004 | 0.008 | 0.008 |
| | 336 | **1.000** | **1.000** | 0.100 | 0.100 | 0.036 | 0.036 | 0.048 | 0.048 | 0.067 | 0.067 | 0.048 | 0.048 | 0.067 | 0.067 |
| | 720 | 0.067 | 0.600 | 0.029 | 0.029 | 0.010 | 0.005 | 0.008 | 0.008 | 0.029 | 0.029 | 0.008 | 0.008 | 0.014 | 0.014 |
| Electricity | 96 | 0.083 | 0.083 | 0.012 | 0.012 | **1.000** | **1.000** | 0.001 | 0.001 | 0.036 | 0.036 | 0.002 | 0.002 | 0.005 | 0.005 |
| | 192 | 0.571 | 0.571 | 0.012 | 0.012 | **1.000** | **1.000** | 0.001 | 0.001 | 0.036 | 0.036 | 0.002 | 0.002 | 0.005 | 0.005 |
| | 336 | **1.000** | **1.000** | 0.100 | 0.100 | 0.679 | 0.679 | 0.048 | 0.048 | 0.167 | 0.167 | 0.048 | 0.048 | 0.067 | 0.067 |
| | 720 | 0.600 | 0.100 | **1.000** | **1.000** | 0.886 | 0.200 | 0.029 | 0.029 | 0.100 | 0.100 | 0.018 | 0.018 | 0.029 | 0.029 |
| Exchange | 96 | 0.038 | 0.020 | 0.152 | 0.001 | 0.000 | 0.000 | **1.000** | **1.000** | 0.001 | 0.001 | 0.000 | 0.000 | 0.000 | 0.000 |
| | 192 | 0.030 | 0.011 | 0.261 | 0.002 | 0.000 | 0.000 | **1.000** | **1.000** | 0.000 | 0.000 | 0.000 | 0.000 | 0.000 | 0.000 |
| | 336 | **1.000** | **1.000** | 0.125 | 0.018 | 0.002 | 0.002 | 0.255 | 0.543 | 0.008 | 0.008 | 0.004 | 0.004 | 0.004 | 0.004 |
| | 720 | **1.000** | **1.000** | 0.029 | 0.029 | 0.005 | 0.005 | 0.000 | 0.000 | 0.014 | 0.014 | 0.008 | 0.008 | 0.008 | 0.008 |
| Traffic | 96 | 0.001 | 0.002 | 0.003 | 0.005 | **1.000** | 0.026 | 0.001 | 0.001 | 0.008 | 0.012 | 0.001 | 0.002 | 0.001 | 0.002 |
| | 192 | 0.005 | **1.000** | 0.012 | 0.029 | **1.000** | 0.129 | 0.001 | 0.005 | 0.036 | 0.067 | 0.002 | 0.008 | 0.002 | 0.008 |
| | 336 | 0.048 | 0.286 | 0.018 | 0.018 | **1.000** | **1.000** | 0.002 | 0.002 | 0.018 | 0.018 | 0.004 | 0.004 | 0.004 | 0.004 |
| | 720 | 0.267 | **1.000** | 0.029 | 0.100 | **1.000** | 0.267 | 0.008 | 0.048 | 0.067 | 0.167 | 0.008 | 0.048 | 0.008 | 0.048 |
| Weather | 96 | **1.000** | **1.000** | 0.029 | 0.029 | 0.003 | 0.003 | 0.002 | 0.002 | 0.018 | 0.018 | 0.012 | 0.012 | 0.012 | 0.012 |
| | 192 | **1.000** | **1.000** | 0.050 | 0.050 | 0.003 | 0.003 | 0.003 | 0.003 | 0.018 | 0.018 | 0.012 | 0.012 | 0.012 | 0.012 |
| | 336 | **1.000** | **1.000** | 0.100 | 0.600 | 0.013 | 0.013 | 0.013 | 0.013 | 0.048 | 0.048 | 0.036 | 0.036 | 0.036 | 0.036 |
| | 720 | **1.000** | 0.100 | 0.400 | **1.000** | 0.013 | 0.003 | 0.028 | 0.008 | 0.048 | 0.018 | 0.036 | 0.012 | 0.036 | 0.012 |

# F. MoF Visualization

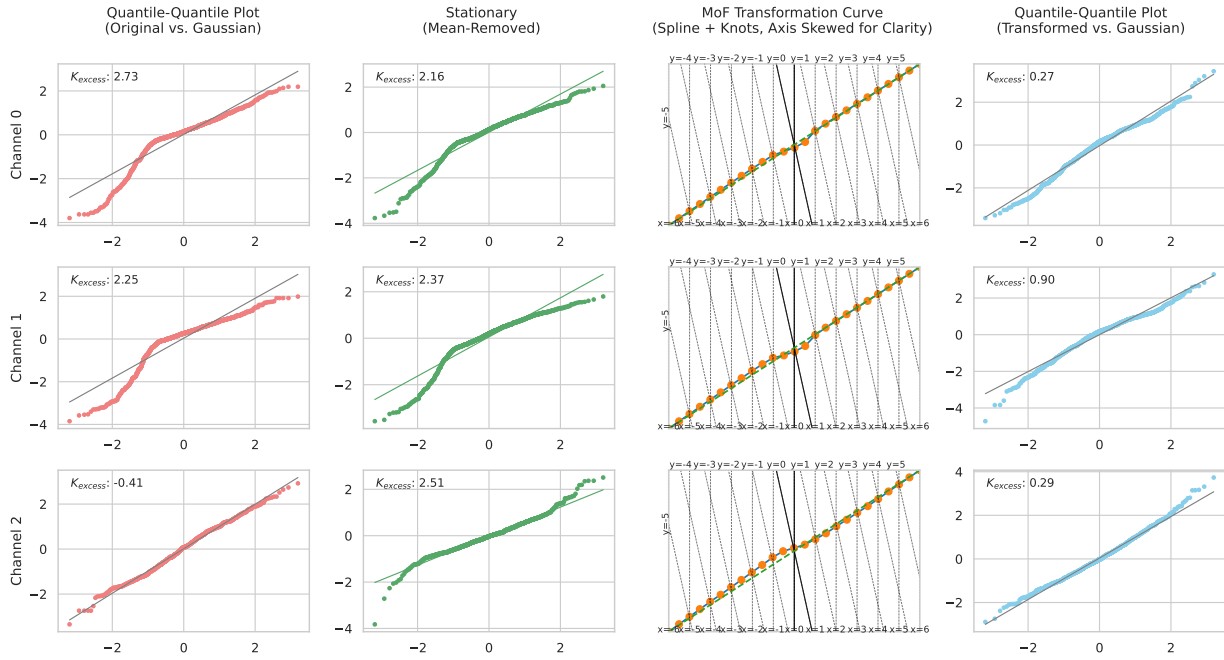

*Figure 13.* Visualization of the transformation process on three seperate dimension of the ETTh1 dataset. After stationarization, the average $K_{excess}$ is 2.34. The MoF transformation reduces the average $K_{excess}$ to 0.49, bringing the data distribution 79.1% closer to normality.

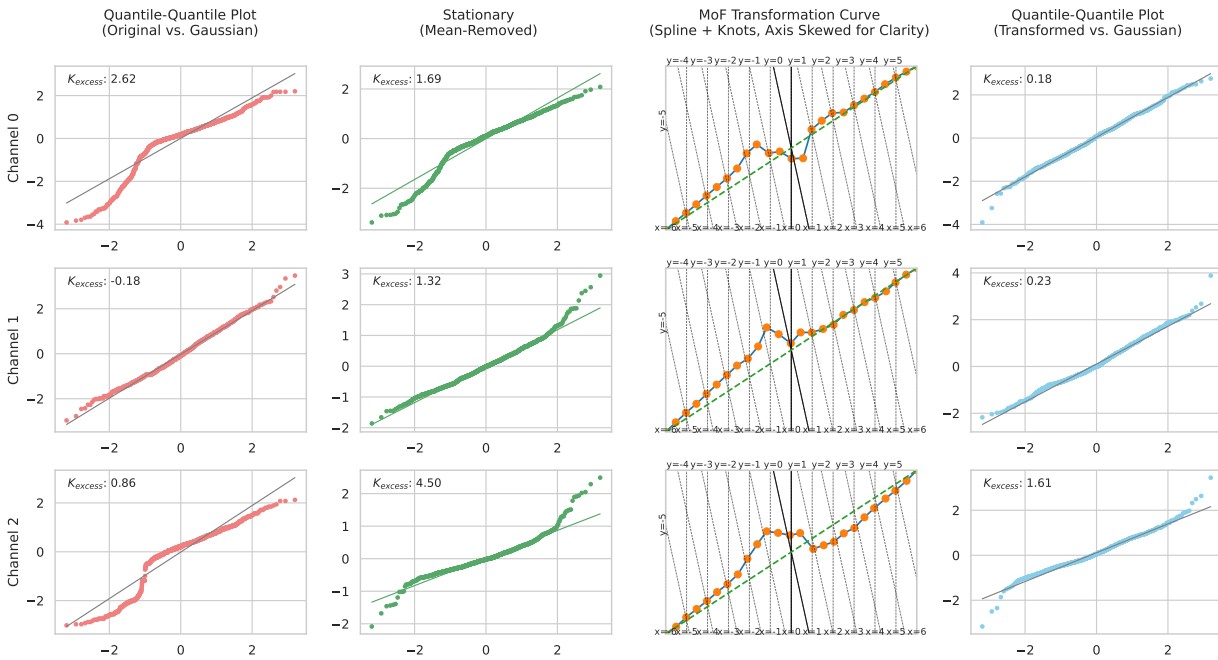

*Figure 14.* Visualization of the transformation process on three seperate dimension of the ETTm1 dataset. After stationarization, the average $K_{excess}$ is 2.50. The MoF transformation reduces the average absolute $K_{excess}$ to 0.67, bringing the data distribution 73.1% closer to normality.

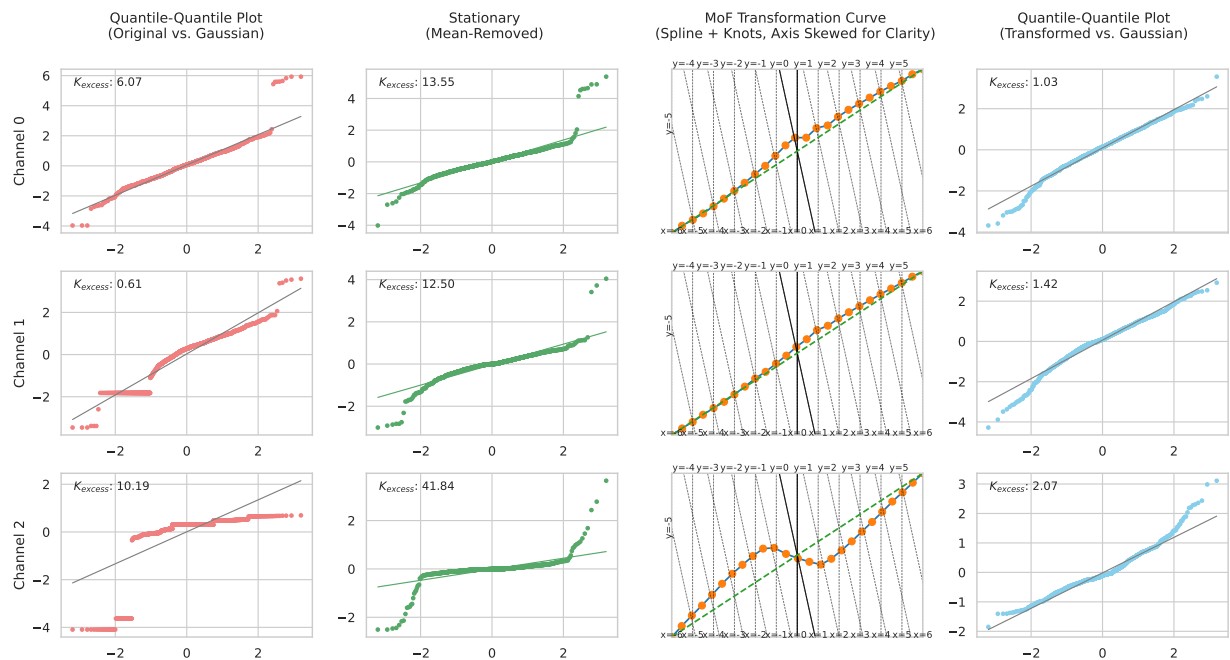

*Figure 15.* Visualization of the transformation process on three seperate dimension of the ETTh2 dataset. After stationarization, the average $K_{excess}$ is 22.63. The MoF transformation reduces the average absolute $K_{excess}$ to 1.50, bringing the data distribution 93.3% closer to normality.

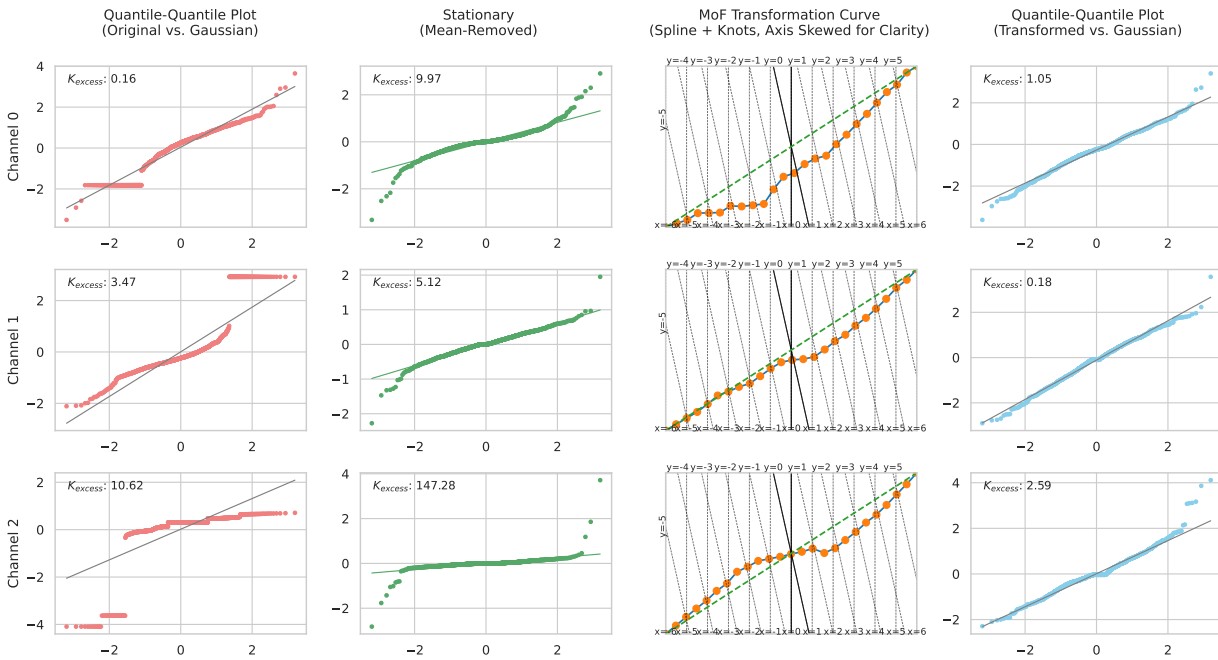

*Figure 16.* Visualization of the transformation process on three seperate dimension of the ETTm2 dataset. After stationarization, the average $K_{excess}$ is 54.12. The MoF transformation reduces the average absolute $K_{excess}$ to 1.27, bringing the data distribution 97.6% closer to normality.

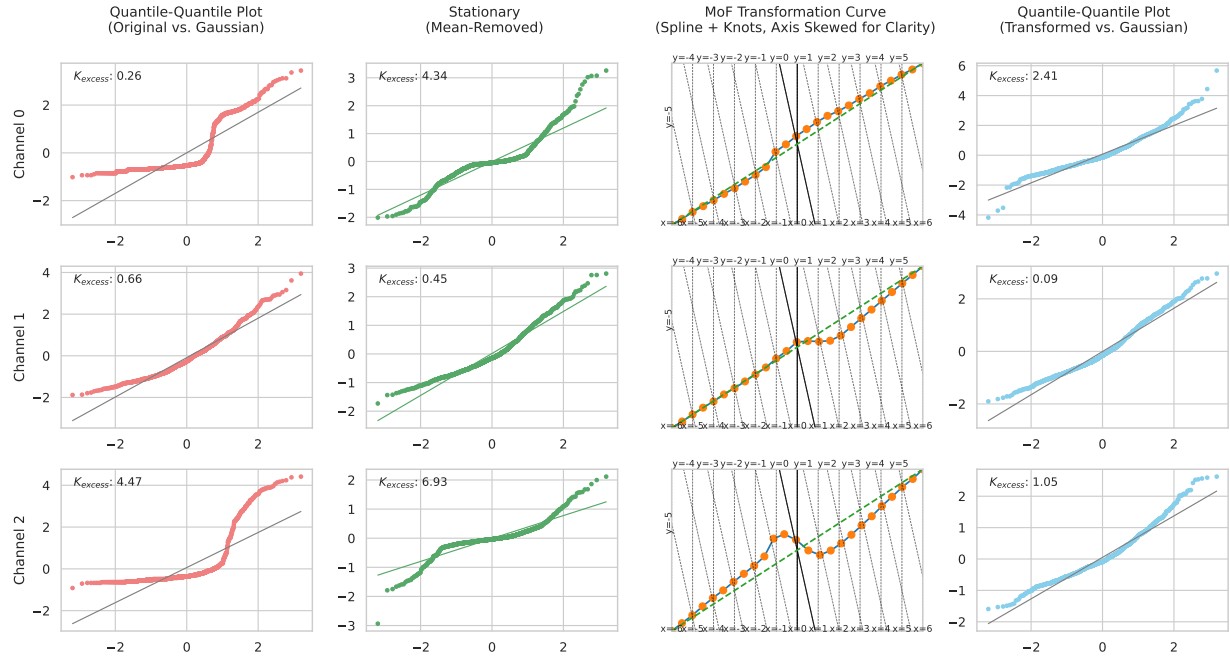

*Figure 17.* Visualization of the transformation process on three separate dimension of the Electricity dataset. After stationarization, the average $K_{excess}$ is 3.91. The MoF transformation reduces the average absolute $K_{excess}$ to 1.18, bringing the data distribution 69.7% closer to normality.

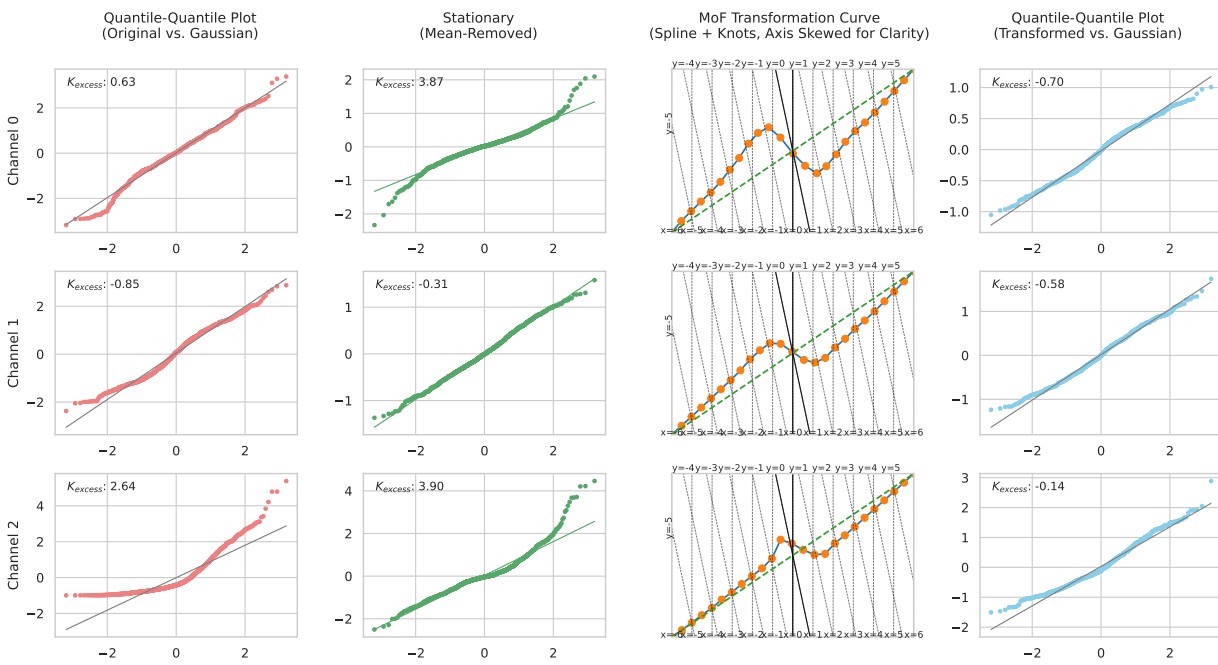

*Figure 18.* Visualization of the transformation process on three separate dimension of the Weather dataset. After stationarization, the average $K_{excess}$ is 2.49. The MoF transformation reduces the average absolute $K_{excess}$ to 0.48, bringing the data distribution 82.3% closer to normality.

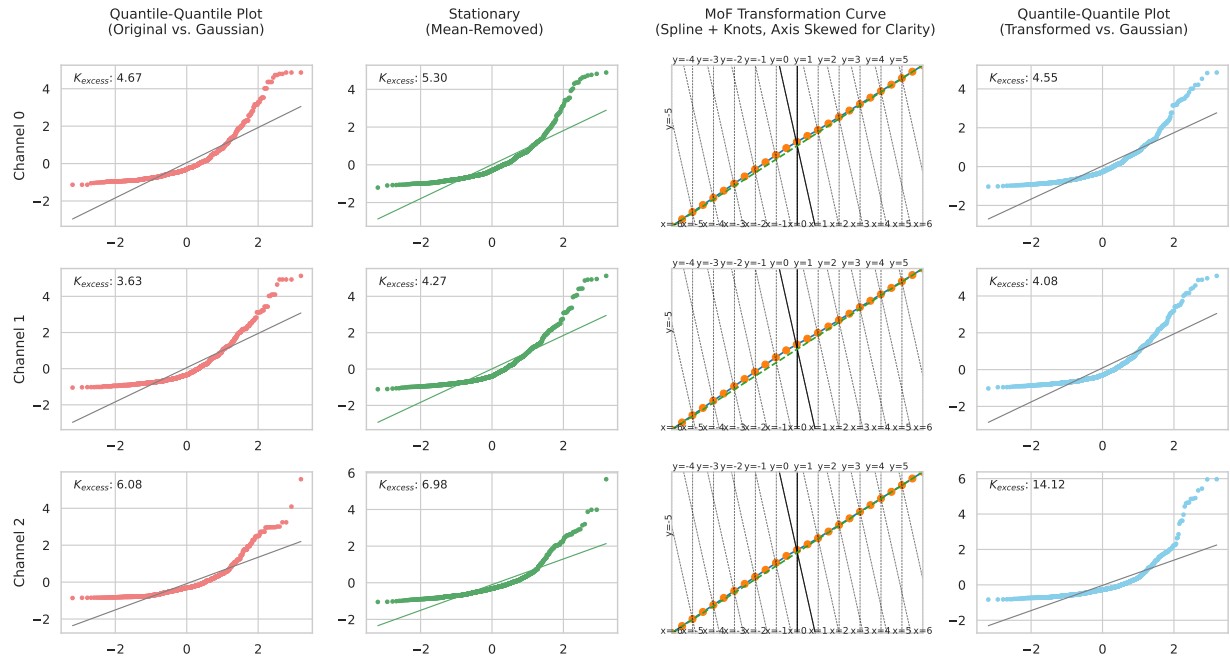

*Figure 19.* Visualization of the transformation process on three separate dimension of the NIL dataset. After stationarization, the average $K_{excess}$ is 5.52. The MoF transformation reduces the average absolute $K_{excess}$ to 7.58, bringing the data distribution 37.4% **further** to normality.Due to limited data size, MoF wasn't able to train sufficiently in this dataset

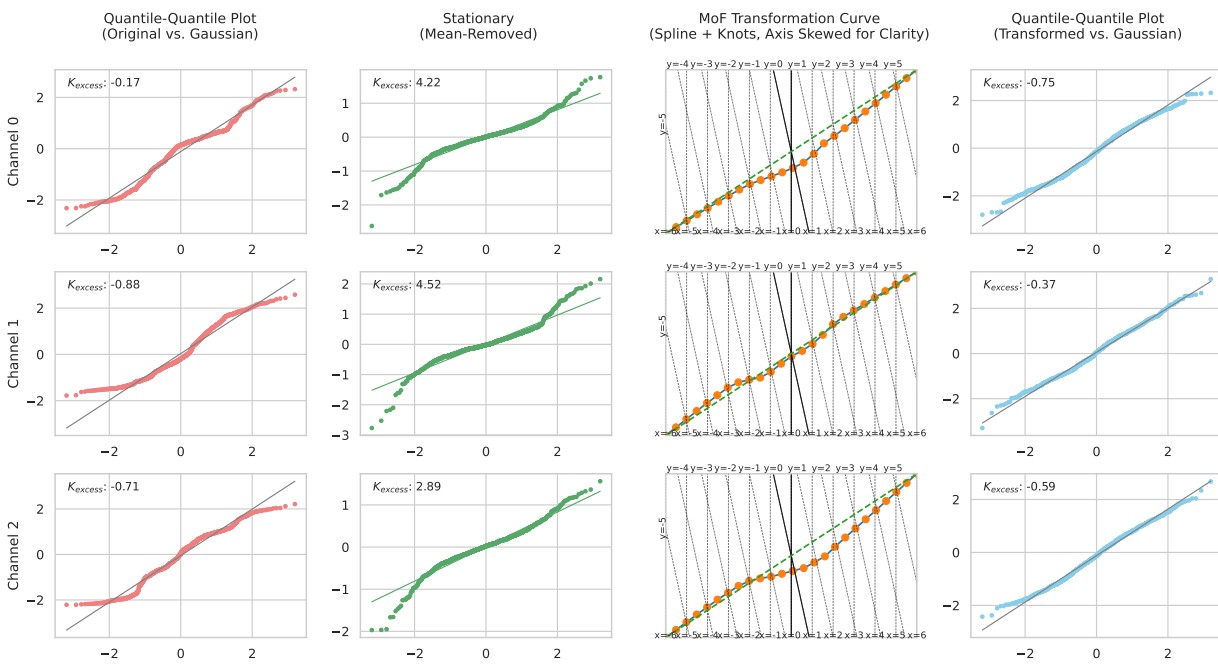

*Figure 20.* Visualization of the transformation process on three separate dimension of the Exchange dataset. After stationarization, the average $K_{excess}$ is 3.88. The MoF transformation reduces the average absolute $K_{excess}$ to 0.57, bringing the data distribution 85.3% closer to normality.

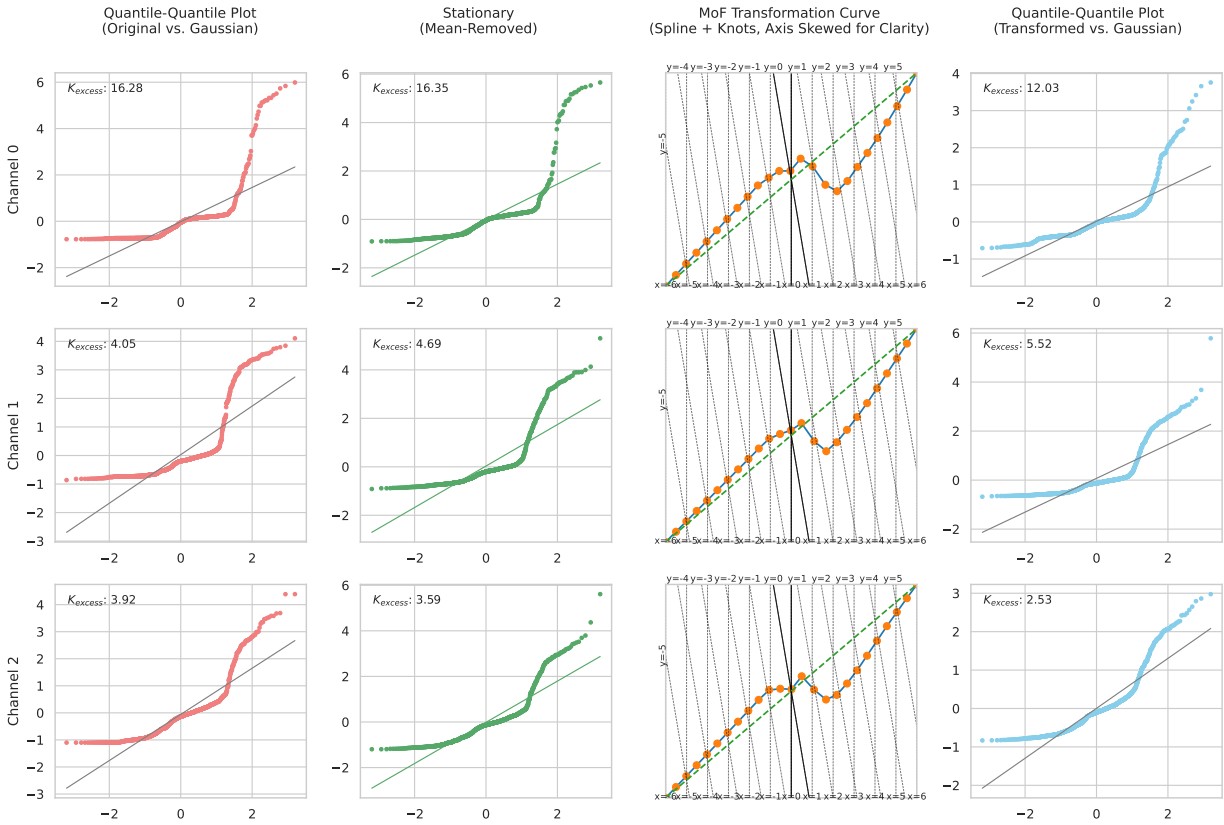

*Figure 21.* Visualization of the transformation process on three separate dimension of the Traffic dataset. After stationarization, the average $K_{excess}$ is 8.21. The MoF transformation reduces the average absolute $K_{excess}$ to 6.69, bringing the data distribution 18.5% closer to normality.

# G. Model Efficiency Comparison

This section provides a detailed explanation of the columns and metrics presented in the table below, comparing our proposed model, MoF, with two backbones (Mamba-64 and Linear), against several state-of-the-art baselines.

- **FLOPs**: The number of floating-point operations in billions, measured using the `fvcore` package. Note that this value is not entirely exclusive and auxiliary computations may occur.

- **Time (ms)**: The running time per 100 iterations in milliseconds, reflecting the practical efficiency of the models.

- **Params**: The total number of trainable parameters in millions, also computed with `fvcore`.

- **Rel.(Relative) MSE and MAE**: The average mean squared error (MSE) and mean absolute error (MAE), computed as follows: each experiment is repeated five times, and the average result is recorded. For comparison, the MSE/MAE is normalized against the best-performing model in each dataset, which typically corresponds to the shortest prediction length (due to lower error).

- **FLOPs Ratio and Param Ratio**: These columns show the ratio of FLOPs and parameter counts relative to MoF with the Linear backbone, serving as a baseline.

- **Perf. Boost**: The percentage improvement or decline in performance compared to the baseline model, MoF with the Linear backbone.

| Model | FLOPs | Time (ms) | Params | Rel. MSE | Rel. MAE | FLOPs Ratio | Param Ratio | Perf. Boost |
|---|---|---|---|---|---|---|---|---|
| **MoF(Mamba-64)** | 1.02E+09 | 1173 | 1.46E+06 | **1.563** | **1.248** | 10.37 | 8.74 | **6.48**% |
| **MoF(Linear)** | **9.86E+07** | 712 | **1.66E+05** | 1.733 | 1.273 | **1.00** | **1.00** | 0.00% |
| Autoformer | 9.22E+09 | 1784 | 6.49E+06 | 3.403 | 1.860 | 93.46 | 39.02 | -75.06% |
| Transformer | 1.10E+10 | 962 | 6.29E+06 | 3.819 | 2.019 | 111.25 | 37.80 | -94.20% |
| Informer | 8.62E+09 | 1269 | 7.08E+06 | 4.862 | 2.263 | 87.38 | 42.53 | -137.05% |
| DLinear | 1.26E+08 | **163** | 3.40E+05 | 1.752 | 1.310 | 1.27 | 2.04 | -1.86% |
| iTransformer | 2.17E+09 | 497 | 3.50E+06 | 1.893 | 1.337 | 22.01 | 21.04 | -7.45% |
| PatchTST | 7.99E+10 | 488 | 1.04E+07 | 1.980 | 1.331 | 810.17 | 62.57 | -10.14% |
| GLinear | 1.26E+08 | 185 | 2.27E+05 | 1.719 | 1.279 | 1.27 | 1.36 | 0.24% |
| CATS | 2.38E+10 | 360 | 1.53E+06 | 2.059 | 1.501 | 241.57 | 9.21 | -18.45% |
| LiNo | 2.60E+09 | 486 | 4.29E+06 | 1.775 | 1.315 | 26.34 | 25.79 | -2.80% |
| TimeMachine | 2.71E+08 | 823 | 8.79E+05 | 1.777 | 1.281 | 2.75 | 5.28 | -1.74% |

**Performance Analysis.** The results demonstrate that our proposed model, MoF, achieves competitive or superior performance compared to state-of-the-art models across various metrics. Notably, MoF consistently delivers strong results in Avg. MSE* and Avg. MAE*, despite having significantly smaller parameter counts. However, it is important to note that the Mamba-64 backbone introduces a binning process that negatively impacts running time, as reflected in the Time (ms) column.

# H. Comprehensive Comparison of MoF with Expanded Input Horizon

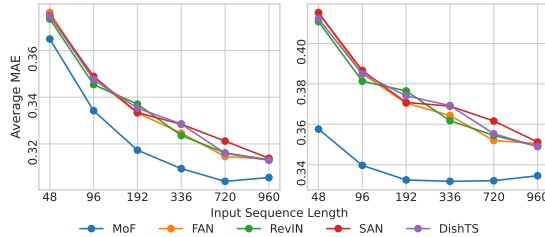

(a) Different normalization method compared on weather dataset with prediction length fixed at 720.

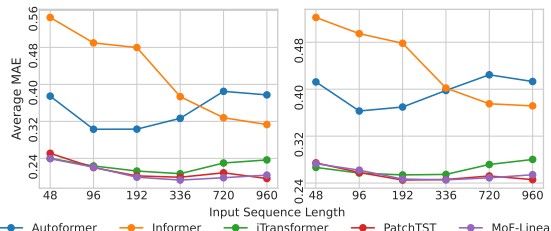

(b) Different backbone method compared on weather dataset with prediction length fixed at 192.

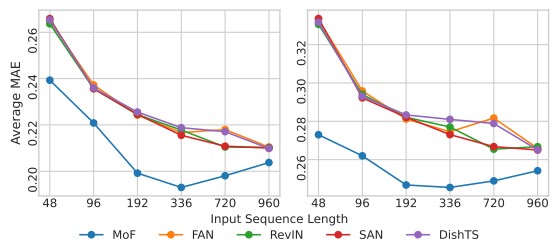

(c) Different normalization method compared on weather dataset with prediction length fixed at 192.

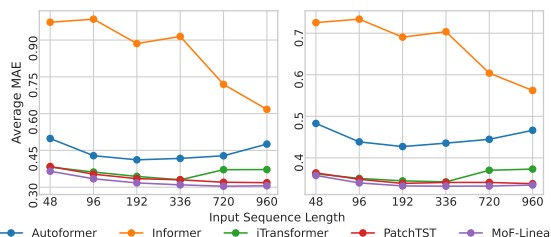

(d) Different backbone method compared on weather dataset with prediction length fixed at 720.

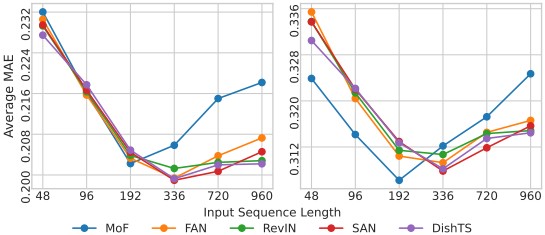

(e) Different normalization method compared on ETTh2 dataset with prediction length fixed at 192.

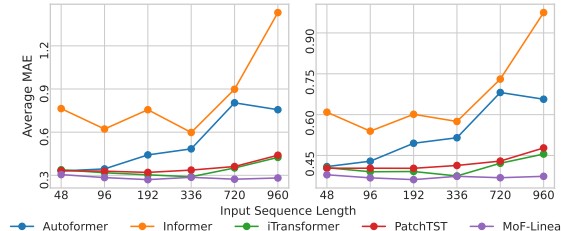

(f) Different backbone method compared on ETTh2 dataset with prediction length fixed at 720.

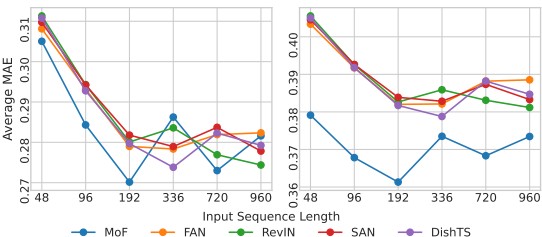

(g) Different normalization method compared on ETTh2 dataset with prediction length fixed at 720.

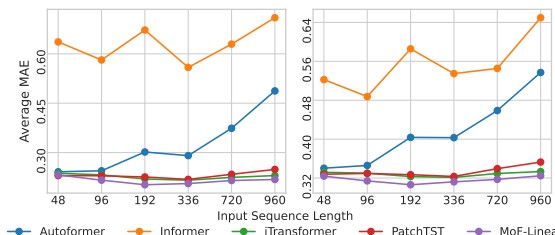

(h) Different backbone method compared on ETTh2 dataset with prediction length fixed at 192.

*Figure 22.* Comparison of different normalization and backbone methods on ETTh2 and weather datasets with prediction lengths fixed at 192 and 720.

# I. Comprehensive Comparison of MoF and Other Normalization Methods

MoF applies simple instance-wise z-score normalization to address non-stationary distributions. All methods are tested with the same parameters across datasets and settings, without dataset-specific tuning.

| Model | | MoF | | RevIN | | FAN | | DishTS | | SAN | | DLinear | |
|---|---|---|---|---|---|---|---|---|---|---|---|---|---|---|
| Dataset | | MSE | MAE | MSE | MAE | MSE | MAE | MSE | MAE | MSE | MAE | MSE | MAE |
| ETTh1 | 96 | 0.415 | **0.438** | 0.418 | 0.439 | **0.414** | 0.445 | 0.421 | 0.444 | 0.426 | 0.448 | 0.435 | 0.444 |
| | 192 | 0.466 | 0.478 | 0.468 | **0.476** | 0.467 | 0.483 | **0.465** | 0.478 | 0.477 | 0.484 | 0.486 | 0.483 |
| | 336 | 0.518 | **0.514** | 0.526 | 0.516 | **0.515** | 0.521 | 0.526 | 0.523 | 0.539 | 0.524 | 0.531 | 0.518 |
| | 720 | 0.657 | 0.602 | 0.667 | 0.604 | **0.634** | 0.598 | 0.656 | 0.608 | 0.676 | 0.608 | 0.644 | **0.594** |
| | Avg. | 0.514 | **0.508** | 0.520 | 0.509 | **0.508** | 0.512 | 0.517 | 0.513 | 0.529 | 0.516 | 0.524 | 0.510 |
| ETTh2 | 96 | 0.165 | 0.278 | 0.165 | **0.277** | 0.173 | 0.288 | **0.165** | 0.279 | 0.172 | 0.285 | 0.176 | 0.285 |
| | 192 | 0.203 | 0.310 | 0.206 | 0.313 | 0.211 | 0.318 | **0.195** | **0.304** | 0.207 | 0.312 | 0.212 | 0.315 |
| | 336 | 0.238 | 0.337 | 0.240 | 0.341 | 0.237 | 0.341 | **0.222** | **0.327** | 0.232 | 0.335 | 0.236 | 0.342 |
| | 720 | 0.281 | 0.371 | 0.317 | 0.394 | 0.279 | 0.382 | **0.261** | **0.356** | 0.288 | 0.386 | 0.286 | 0.386 |
| | Avg. | 0.222 | 0.324 | 0.232 | 0.331 | 0.225 | 0.332 | **0.211** | **0.316** | 0.225 | 0.329 | 0.228 | 0.332 |
| ETTm1 | 96 | 0.328 | 0.379 | 0.333 | 0.373 | **0.319** | 0.370 | 0.325 | **0.368** | 0.328 | 0.384 | 0.355 | 0.385 |
| | 192 | 0.376 | 0.412 | 0.385 | 0.404 | **0.368** | 0.404 | 0.377 | **0.400** | 0.373 | 0.412 | 0.420 | 0.419 |
| | 336 | 0.418 | 0.441 | 0.435 | 0.436 | **0.416** | 0.438 | 0.422 | **0.432** | 0.418 | 0.447 | 0.484 | 0.455 |
| | 720 | 0.478 | 0.485 | 0.497 | 0.479 | **0.467** | 0.479 | 0.478 | **0.473** | 0.474 | 0.483 | 0.547 | 0.501 |
| | Avg. | 0.400 | 0.429 | 0.412 | 0.423 | **0.393** | 0.423 | 0.401 | **0.418** | 0.398 | 0.432 | 0.452 | 0.440 |
| ETTm2 | 96 | 0.118 | 0.228 | 0.113 | 0.226 | 0.117 | 0.230 | **0.112** | 0.225 | 0.114 | **0.224** | 0.120 | 0.234 |
| | 192 | 0.164 | 0.263 | 0.145 | 0.256 | 0.148 | 0.259 | **0.140** | **0.251** | 0.148 | 0.256 | 0.148 | 0.262 |
| | 336 | 0.200 | 0.292 | 0.171 | 0.277 | 0.181 | 0.287 | **0.169** | **0.277** | 0.175 | 0.282 | 0.180 | 0.292 |
| | 720 | 0.248 | 0.329 | 0.219 | 0.315 | 0.219 | 0.321 | **0.210** | **0.309** | 0.215 | 0.317 | 0.232 | 0.333 |
| | Avg. | 0.182 | 0.278 | 0.162 | 0.268 | 0.166 | 0.274 | **0.158** | **0.265** | 0.163 | 0.270 | 0.170 | 0.280 |
| Electricity | 96 | 0.137 | 0.232 | 0.140 | 0.236 | 0.139 | 0.238 | 0.138 | 0.235 | **0.134** | **0.231** | 0.140 | 0.237 |
| | 192 | 0.151 | **0.245** | 0.154 | 0.248 | 0.156 | 0.255 | 0.154 | 0.250 | **0.150** | 0.246 | 0.153 | 0.250 |
| | 336 | 0.166 | **0.261** | 0.171 | 0.265 | 0.174 | 0.274 | 0.170 | 0.268 | **0.166** | 0.265 | 0.169 | 0.268 |
| | 720 | 0.201 | **0.293** | 0.210 | 0.297 | 0.212 | 0.310 | 0.207 | 0.303 | **0.201** | 0.298 | 0.203 | 0.301 |
| | Avg. | 0.164 | **0.258** | 0.169 | 0.261 | 0.170 | 0.269 | 0.167 | 0.264 | **0.163** | 0.260 | 0.166 | 0.264 |
| Exchange | 96 | 0.092 | 0.211 | 0.090 | 0.210 | 0.136 | 0.274 | 0.126 | 0.278 | 0.123 | 0.249 | **0.084** | **0.209** |
| | 192 | 0.194 | 0.310 | 0.188 | 0.307 | 0.252 | 0.374 | 0.268 | 0.389 | 0.193 | 0.323 | **0.171** | **0.306** |
| | 336 | 0.363 | 0.433 | 0.346 | **0.425** | 0.594 | 0.589 | 0.445 | 0.545 | 0.353 | 0.440 | **0.332** | 0.435 |
| | 720 | 0.958 | 0.728 | **0.915** | **0.715** | 1.306 | 0.847 | 1.904 | 1.084 | 1.201 | 0.775 | 0.970 | 0.742 |
| | Avg. | 0.401 | 0.421 | **0.385** | **0.414** | 0.572 | 0.521 | 0.686 | 0.574 | 0.467 | 0.447 | 0.389 | 0.423 |
| Traffic | 96 | **0.407** | **0.279** | 0.411 | 0.281 | 0.411 | 0.290 | 0.423 | 0.295 | 0.417 | 0.283 | 0.411 | 0.284 |
| | 192 | **0.420** | **0.283** | 0.424 | 0.286 | 0.428 | 0.300 | 0.437 | 0.302 | 0.435 | 0.290 | 0.423 | 0.290 |
| | 336 | **0.431** | **0.291** | 0.437 | 0.293 | 0.446 | 0.312 | 0.451 | 0.310 | 0.451 | 0.298 | 0.436 | 0.297 |
| | 720 | **0.462** | 0.313 | 0.465 | **0.309** | 0.481 | 0.333 | 0.482 | 0.329 | 0.481 | 0.314 | 0.466 | 0.317 |
| | Avg. | **0.430** | **0.291** | 0.434 | 0.292 | 0.441 | 0.309 | 0.448 | 0.309 | 0.446 | 0.296 | 0.434 | 0.297 |
| Weather | 96 | **0.148** | **0.202** | 0.173 | 0.222 | 0.153 | 0.210 | 0.166 | 0.240 | 0.146 | 0.208 | 0.174 | 0.234 |
| | 192 | **0.193** | **0.244** | 0.216 | 0.258 | 0.197 | 0.257 | 0.207 | 0.278 | 0.192 | 0.254 | 0.216 | 0.276 |
| | 336 | **0.239** | **0.281** | 0.263 | 0.292 | 0.246 | 0.294 | 0.258 | 0.323 | 0.241 | 0.292 | 0.263 | 0.315 |
| | 720 | **0.308** | **0.333** | 0.331 | 0.339 | 0.314 | 0.344 | 0.330 | 0.382 | 0.318 | 0.348 | 0.324 | 0.363 |
| | Avg. | **0.222** | **0.265** | 0.246 | 0.278 | 0.228 | 0.276 | 0.240 | 0.305 | 0.224 | 0.275 | 0.244 | 0.297 |
| Avg.(ALL) | | **0.317** | **0.347** | 0.320 | 0.348 | 0.338 | 0.365 | 0.371 | 0.354 | 0.327 | 0.354 | 0.356 | 0.326 |
| Diff.(%) | | **0.00** | **0.00** | -0.94 | -0.12 | -6.55 | -5.15 | -6.93 | -11.48 | -3.15 | -1.86 | -2.50 | -2.81 |

# J. Module generalization test

| Model | Autoformer | | | | Informer | | | | DLinear | | | |
| --- | --- | --- | --- | --- | --- | --- | --- | --- | --- | --- | --- | --- |
| | w/o MoF | | w/ MoF | | w/o MoF | | w/ MoF | | w/o MoF | | w/ MoF | |
| Dataset | MAE | MSE | MAE | MSE | MAE | MSE | MAE | MSE | MAE | MSE | MAE | MSE |
| ETTh1 96 | 0.596 | 0.649 | **0.568** | **0.631** | 0.834 | 1.226 | **0.606** | **0.716** | 0.444 | 0.436 | **0.439** | **0.416** |
| ETTh1 192 | 0.635 | 0.735 | **0.576** | **0.632** | 0.867 | 1.328 | **0.648** | **0.782** | 0.483 | 0.486 | **0.479** | **0.467** |
| ETTh1 336 | 0.690 | 0.843 | **0.623** | **0.734** | 0.786 | 1.121 | **0.663** | **0.797** | 0.518 | 0.531 | **0.514** | **0.519** |
| ETTh1 720 | 0.705 | 0.857 | **0.715** | **0.875** | 0.864 | 1.295 | **0.763** | **1.096** | 0.594 | 0.645 | **0.603** | **0.658** |
| ETTh1 Avg. | 0.656 | 0.771 | **0.620** | **0.718** | 0.838 | 1.243 | **0.670** | **0.848** | 0.510 | 0.524 | **0.509** | **0.515** |
| ETTh2 96 | 0.367 | 0.248 | **0.357** | **0.255** | 0.440 | 0.362 | **0.372** | **0.291** | 0.285 | 0.176 | **0.279** | **0.166** |
| ETTh2 192 | 0.407 | 0.302 | **0.380** | **0.275** | 0.484 | 0.439 | **0.400** | **0.319** | 0.316 | 0.213 | **0.310** | **0.204** |
| ETTh2 336 | 0.442 | 0.349 | **0.389** | **0.302** | 0.478 | 0.434 | **0.434** | **0.382** | 0.342 | **0.236** | **0.337** | 0.238 |
| ETTh2 720 | 0.566 | 0.598 | **0.410** | **0.322** | 0.578 | 0.599 | **0.417** | **0.352** | 0.386 | 0.287 | **0.371** | **0.282** |
| ETTh2 Avg. | 0.446 | 0.374 | **0.384** | **0.289** | 0.495 | 0.458 | **0.406** | **0.336** | 0.332 | 0.228 | **0.324** | **0.222** |
| ETTm1 96 | 0.559 | 0.617 | **0.533** | **0.570** | 0.558 | 0.621 | **0.493** | **0.558** | 0.386 | 0.355 | **0.379** | **0.328** |
| ETTm1 192 | **0.556** | **0.583** | 0.557 | 0.636 | 0.609 | 0.703 | **0.521** | **0.608** | 0.420 | 0.421 | **0.412** | **0.377** |
| ETTm1 336 | 0.569 | 0.609 | **0.550** | **0.595** | 0.658 | 0.783 | **0.558** | **0.631** | 0.456 | 0.485 | **0.441** | **0.419** |
| ETTm1 720 | 0.606 | 0.668 | **0.576** | **0.631** | 0.706 | 0.872 | **0.577** | **0.687** | 0.502 | 0.547 | **0.486** | **0.479** |
| ETTm1 Avg. | 0.573 | 0.619 | **0.554** | **0.608** | 0.633 | 0.745 | **0.537** | **0.621** | 0.441 | 0.452 | **0.429** | **0.401** |
| ETTm2 96 | **0.283** | **0.163** | 0.285 | 0.167 | 0.308 | 0.208 | **0.264** | **0.153** | 0.234 | 0.120 | **0.228** | **0.118** |
| ETTm2 192 | **0.298** | **0.181** | 0.304 | 0.196 | 0.367 | 0.292 | **0.310** | **0.208** | **0.263** | **0.148** | 0.263 | 0.165 |
| ETTm2 336 | **0.315** | **0.205** | 0.323 | 0.219 | 0.404 | 0.350 | **0.336** | **0.247** | **0.292** | **0.181** | 0.292 | 0.200 |
| ETTm2 720 | 0.347 | **0.247** | **0.346** | 0.248 | 0.550 | 0.672 | **0.377** | **0.306** | 0.333 | **0.232** | **0.329** | 0.248 |
| ETTm2 Avg. | **0.311** | **0.199** | 0.315 | 0.207 | 0.407 | 0.381 | **0.322** | **0.228** | 0.280 | **0.170** | **0.278** | 0.183 |
| Electricity 96 | 0.316 | 0.203 | **0.293** | **0.192** | 0.418 | 0.344 | **0.311** | **0.211** | 0.238 | 0.140 | **0.233** | **0.137** |
| Electricity 192 | 0.329 | 0.213 | **0.296** | **0.192** | 0.433 | 0.364 | **0.323** | **0.226** | 0.251 | 0.154 | **0.246** | **0.151** |
| Electricity 336 | 0.334 | 0.219 | **0.302** | **0.196** | 0.433 | 0.367 | **0.329** | **0.231** | 0.268 | 0.169 | **0.262** | **0.167** |
| Electricity 720 | 0.352 | 0.247 | **0.319** | **0.219** | 0.415 | 0.342 | **0.358** | **0.271** | 0.301 | 0.204 | **0.293** | **0.201** |
| Electricity Avg. | 0.333 | 0.221 | **0.302** | **0.200** | 0.425 | 0.354 | **0.330** | **0.235** | 0.265 | 0.167 | **0.259** | **0.164** |
| Exchange 96 | 0.629 | 0.660 | **0.476** | **0.424** | 0.898 | 1.159 | **0.403** | **0.307** | **0.209** | **0.084** | 0.211 | 0.092 |
| Exchange 192 | 0.900 | 1.274 | **0.494** | **0.456** | 0.973 | 1.356 | **0.510** | **0.523** | **0.307** | **0.171** | 0.310 | 0.194 |
| Exchange 336 | 0.916 | 1.313 | **0.599** | **0.667** | 1.049 | 1.635 | **0.642** | **0.733** | 0.435 | **0.332** | **0.434** | 0.363 |
| Exchange 720 | 0.947 | 1.547 | **0.828** | **1.213** | 1.315 | 2.553 | **0.789** | **1.212** | 0.743 | 0.971 | **0.729** | **0.958** |
| Exchange Avg. | 0.848 | 1.198 | **0.599** | **0.690** | 1.059 | 1.676 | **0.586** | **0.694** | 0.423 | **0.390** | **0.421** | 0.402 |
| NIL 24 | **1.338** | 3.716 | 1.377 | **3.646** | 1.518 | 4.964 | **1.166** | **3.173** | 1.322 | 3.554 | **1.072** | **2.623** |
| NIL 36 | **1.235** | 3.273 | 1.373 | 3.701 | 1.495 | 4.816 | **1.220** | **3.584** | 1.336 | 3.680 | **1.053** | **2.567** |
| NIL 48 | **1.212** | **3.244** | 1.322 | 3.597 | 1.522 | 4.875 | **1.190** | **3.349** | 1.356 | 3.808 | **1.076** | **2.660** |
| NIL 60 | 1.200 | 3.290 | **1.167** | **3.003** | 1.610 | 5.397 | **1.252** | **3.714** | 1.413 | 4.206 | **1.099** | **2.780** |
| NIL Avg. | **1.246** | **3.381** | 1.310 | 3.487 | 1.536 | 5.013 | **1.207** | **3.455** | 1.357 | 3.812 | **1.075** | **2.658** |
| Traffic 96 | 0.384 | **0.637** | **0.367** | 0.671 | 0.445 | 0.793 | **0.402** | **0.723** | 0.285 | 0.411 | **0.279** | **0.407** |
| Traffic 192 | 0.397 | **0.660** | **0.379** | 0.692 | 0.416 | 0.744 | **0.382** | **0.691** | 0.290 | 0.424 | **0.284** | **0.420** |
| Traffic 336 | 0.377 | **0.628** | **0.364** | 0.672 | 0.488 | 0.894 | **0.413** | **0.747** | 0.298 | 0.437 | **0.291** | **0.432** |
| Traffic 720 | 0.406 | **0.668** | **0.369** | 0.674 | 0.450 | 0.815 | **0.441** | **0.804** | 0.317 | 0.467 | **0.313** | **0.463** |
| Traffic Avg. | 0.391 | **0.648** | **0.370** | 0.677 | 0.450 | 0.811 | **0.410** | **0.741** | 0.297 | 0.435 | **0.292** | **0.430** |
| Weather 96 | 0.365 | 0.290 | **0.290** | **0.246** | 0.357 | 0.290 | **0.273** | **0.224** | 0.235 | 0.174 | **0.202** | **0.149** |
| Weather 192 | 0.399 | 0.342 | **0.331** | **0.305** | 0.400 | 0.356 | **0.327** | **0.346** | 0.276 | 0.217 | **0.245** | **0.193** |
| Weather 336 | 0.422 | 0.376 | **0.343** | **0.321** | 0.563 | 0.621 | **0.360** | **0.435** | 0.316 | 0.263 | **0.282** | **0.239** |
| Weather 720 | 0.445 | 0.438 | **0.368** | **0.380** | 0.666 | 0.850 | **0.385** | **0.420** | 0.363 | 0.325 | **0.333** | **0.308** |
| Weather Avg. | 0.408 | 0.361 | **0.333** | **0.313** | 0.496 | 0.529 | **0.336** | **0.356** | 0.297 | 0.245 | **0.266** | **0.222** |

## K. Comparison with IN-Flow and FITS

| | Model Len | MoF-Linear | | MoF-Mamba | | INFlow-Linear | | INFlow-Patch | | FITS | |
|---|---|---|---|---|---|---|---|---|---|---|---|
| | | MAE | MSE | MAE | MSE | MAE | MSE | MAE | MSE | MAE | MSE |
| ETTh1 | 96 | 0.441 | **0.406** | **0.439** | 0.416 | 0.440 | 0.417 | 0.451 | 0.421 | 0.682 | 0.909 |
| | 192 | 0.480 | **0.458** | 0.479 | 0.467 | **0.477** | 0.468 | 0.489 | 0.482 | 0.696 | 0.937 |
| | 336 | 0.522 | **0.513** | **0.514** | 0.519 | 0.518 | 0.528 | 0.535 | 0.550 | 0.718 | 0.979 |
| | 720 | **0.595** | **0.627** | 0.603 | 0.658 | 0.607 | 0.670 | 0.633 | 0.709 | 0.777 | 1.096 |
| | Avg. | 0.510 | **0.501** | **0.509** | 0.515 | 0.510 | 0.521 | 0.527 | 0.540 | 0.718 | 0.980 |
| ETTh2 | 96 | 0.286 | 0.173 | 0.279 | 0.166 | **0.278** | **0.165** | 0.292 | 0.181 | 0.350 | 0.246 |
| | 192 | 0.314 | 0.207 | **0.310** | 0.204 | 0.311 | **0.203** | 0.325 | 0.217 | 0.366 | 0.268 |
| | 336 | 0.338 | **0.236** | 0.337 | 0.238 | 0.343 | 0.243 | 0.358 | 0.263 | 0.385 | 0.295 |
| | 720 | 0.394 | **0.281** | **0.371** | 0.282 | 0.394 | 0.318 | 0.414 | 0.337 | 0.418 | 0.348 |
| | Avg. | 0.333 | 0.224 | **0.324** | **0.222** | 0.332 | 0.232 | 0.347 | 0.249 | 0.380 | 0.289 |
| ETTm1 | 96 | 0.376 | **0.324** | 0.379 | 0.328 | **0.372** | 0.331 | 0.383 | 0.335 | 0.649 | 0.852 |
| | 192 | **0.402** | 0.368 | 0.412 | 0.377 | 0.405 | 0.385 | 0.419 | 0.383 | 0.660 | 0.871 |
| | 336 | **0.433** | 0.407 | 0.441 | 0.419 | 0.436 | 0.434 | 0.447 | 0.421 | 0.676 | 0.898 |
| | 720 | 0.484 | **0.468** | 0.486 | 0.479 | **0.480** | 0.497 | 0.494 | 0.487 | 0.696 | 0.936 |
| | Avg. | 0.424 | **0.392** | 0.429 | 0.401 | **0.423** | 0.412 | 0.436 | 0.406 | 0.670 | 0.889 |
| ETTm2 | 96 | **0.226** | **0.113** | 0.228 | 0.118 | 0.227 | 0.114 | **0.226** | **0.113** | 0.296 | 0.176 |
| | 192 | 0.256 | 0.146 | 0.263 | 0.165 | **0.255** | **0.144** | 0.258 | 0.146 | 0.309 | 0.194 |
| | 336 | 0.282 | 0.175 | 0.292 | 0.200 | **0.279** | **0.173** | 0.288 | 0.177 | 0.326 | 0.220 |
| | 720 | 0.326 | 0.227 | 0.329 | 0.248 | **0.316** | **0.220** | 0.330 | 0.235 | 0.352 | 0.259 |
| | Avg. | 0.273 | 0.165 | 0.278 | 0.183 | **0.269** | **0.163** | 0.276 | 0.168 | 0.321 | 0.212 |
| Electricity | 96 | **0.228** | **0.132** | 0.233 | 0.137 | 0.248 | 0.149 | 0.271 | 0.172 | 0.757 | 0.835 |
| | 192 | **0.242** | **0.149** | 0.246 | 0.151 | 0.262 | 0.165 | 0.299 | 0.200 | 0.760 | 0.845 |
| | 336 | **0.259** | **0.165** | 0.262 | 0.167 | 0.275 | 0.178 | 0.494 | 0.454 | 0.764 | 0.857 |
| | 720 | 0.294 | 0.205 | **0.293** | **0.201** | 0.305 | 0.217 | 0.614 | 0.652 | 0.773 | 0.883 |
| | Avg. | **0.256** | **0.163** | 0.259 | 0.164 | 0.273 | 0.177 | 0.419 | 0.370 | 0.763 | 0.855 |
| Exchange | 96 | 0.214 | **0.088** | **0.211** | 0.092 | 0.212 | 0.093 | **0.211** | 0.090 | 0.416 | 0.307 |
| | 192 | 0.315 | **0.186** | 0.310 | 0.194 | **0.309** | 0.190 | 0.328 | 0.208 | 0.483 | 0.413 |
| | 336 | **0.426** | **0.325** | 0.434 | 0.363 | 0.435 | 0.368 | 0.431 | 0.360 | 0.581 | 0.577 |
| | 720 | **0.621** | **0.621** | 0.729 | 0.958 | 0.724 | 0.947 | 0.745 | 1.020 | 0.813 | 1.143 |
| | Avg. | **0.394** | **0.305** | 0.421 | 0.402 | 0.420 | 0.399 | 0.429 | 0.419 | 0.573 | 0.610 |
| NIL | 24 | 1.077 | **2.508** | **1.072** | 2.623 | 1.128 | 2.808 | 1.077 | 2.597 | 1.570 | 4.865 |
| | 36 | 1.055 | **2.439** | **1.053** | 2.567 | 1.109 | 2.774 | 1.068 | 2.570 | 1.568 | 4.913 |
| | 48 | **1.066** | **2.529** | 1.076 | 2.660 | 1.119 | 2.801 | 1.101 | 2.714 | 1.574 | 5.012 |
| | 60 | **1.079** | **2.615** | 1.099 | 2.780 | 1.123 | 2.843 | 1.116 | 2.729 | 1.603 | 5.216 |
| | Avg. | **1.069** | **2.523** | 1.075 | 2.658 | 1.120 | 2.806 | 1.091 | 2.652 | 1.579 | 5.002 |
| Traffic | 96 | **0.271** | **0.391** | 0.279 | 0.407 | 0.301 | 0.429 | 0.687 | 1.508 | 0.794 | 1.378 |
| | 192 | **0.277** | **0.407** | 0.284 | 0.420 | 0.308 | 0.448 | 0.308 | 0.445 | 0.798 | 1.393 |
| | 336 | **0.284** | **0.420** | 0.291 | 0.432 | 0.315 | 0.459 | 0.396 | 0.566 | 0.800 | 1.408 |
| | 720 | **0.300** | **0.447** | 0.313 | 0.463 | 0.332 | 0.489 | 0.597 | 1.780 | 0.804 | 1.428 |
| | Avg. | **0.283** | **0.416** | 0.292 | 0.430 | 0.314 | 0.456 | 0.497 | 1.075 | 0.799 | 1.402 |
| Weather | 96 | **0.196** | **0.145** | 0.202 | 0.149 | 0.218 | 0.166 | 0.201 | 0.153 | 0.311 | 0.256 |
| | 192 | **0.237** | **0.186** | 0.245 | 0.193 | 0.254 | 0.209 | 0.244 | 0.198 | 0.328 | 0.284 |
| | 336 | **0.278** | **0.238** | 0.282 | 0.239 | 0.291 | 0.256 | 0.284 | 0.251 | 0.347 | 0.317 |
| | 720 | **0.332** | 0.317 | 0.333 | **0.308** | 0.340 | 0.327 | 0.338 | 0.324 | 0.380 | 0.368 |
| | Avg. | **0.261** | **0.221** | 0.266 | 0.222 | 0.276 | 0.239 | 0.267 | 0.231 | 0.342 | 0.306 |

*Table 11.* Comparison with IN-Flow and FITS(Best in **bold**). IN-Flow focuses on nonstationarity (not fat-tail), showing decent performance on smaller ETT sets but struggling with large-channel data (Electricity, Traffic). FITS is parameter-efficient but relies on extensive hyperparameter tuning and is unstable under a unified setting.

## L. Temporal Complexity Analysis

In this section, we analyze the time (temporal) complexity of the two core components of our proposed *MoF* framework: the *Flow* layer (Section 3.1) and the *Morph* module (Section 3.2). We focus on the key (hyper)parameters and how they affect runtime with respect to the sequence length $T$, the number of channels $C$, and the number of bins $B$ used in our piecewise-linear spline models.

### L.1. Flow Layer Complexity

Recall that the Flow layer (cf. Section 3.1) applies a piecewise-linear spline transform to each input channel. Each channel $c \in \{1, \ldots, C\}$ has $B$ bins, and each bin is defined by a width $\{w_{c,i}\}$ and a height $\{h_{c,i}\}$, where $i \in \{1, \ldots, B\}$.

**Forward pass.** In the forward direction, every input element $x_{c,t}$ for channel $c$ and time step $t \in \{1, \ldots, T\}$ is mapped to some output $y_{c,t}$. To find the appropriate bin for $x_{c,t}$, one can:

1. Perform a (potentially) linear scan or prefix-sum-based search among the $B$ bins.

2. Compute the output via one linear operation within the chosen bin.

A direct (linear) scan to select a bin requires $\mathcal{O}(B)$ time, and this is repeated for each channel ($C$) and time step ($T$). Thus, the worst-case complexity for the forward pass across all elements is:

$$\mathcal{O}\big(C \times T \times B\big).$$

If a more efficient bin-search strategy is employed (e.g., binary search), the factor $B$ could be reduced to $\log B$. However, we assume the simpler linear approach here, yielding the $\mathcal{O}(C\,T\,B)$ cost.

**Inverse pass.** The inverse transformation proceeds analogously: one locates the bin for each output element $y_{c,t}$ in $\mathcal{O}(B)$ and then applies a linear mapping to recover $x_{c,t}$. As above, this yields:

$$\mathcal{O}\big(C \times T \times B\big)$$

time complexity for the inverse pass. Since both forward and inverse passes are required for training end-to-end (and for any invertible steps within the backbone), the overall complexity for the Flow layer remains $\mathcal{O}(C\,T\,B)$.

### L.2. Morph Module Complexity

The Morph module (cf. Section 3.2) consists of two main parts:

1. The *test-time-trained temporal layer*, which optimizes a low-rank weight matrix $\mathbf{W}_{\text{test}}$ using a self-supervised loss.

2. The *Up-Projection* step that uses the updated $\mathbf{W}_{\text{test}}$ to generate the scaling factor $\mathbf{x}_{\text{mod}}$, then re-applies the Flow transformation with modified parameters.

**Test-time training.** During inference, each incoming batch (or sequence) of length $T$ and channel dimension $C$ is first transformed by the Flow layer into $\mathbf{x}'$. The Morph module then:

- Projects $\mathbf{x}'$ into query and key embeddings , incurring $\mathcal{O}(C\,T\,d)$ costs if these linear projections map $\mathbf{x}'$ into an internal dimension $d$.

- Computes a self-supervised loss $\ell(\mathbf{W}_{\text{test}}; \mathbf{x}')$ in $\mathcal{O}(C\,T)$ time (since it involves elementwise differences under a binary mask $\mathbf{M}$).

- Performs one or more gradient steps to update $\mathbf{W}_{\text{test}}$. Each gradient step requires backpropagation through the above projections, adding another $\mathcal{O}(C\,T\,d)$ factor per update.

If we denote by $N_{\text{iter}}$ the number of gradient steps performed per test batch, the test-time training complexity scales as:

$$\mathcal{O}\big(N_{\text{iter}} \times (C\,T\,d)\big).$$

In practice, $N_{\text{iter}}$ is typically small (often 1 to 5) to maintain real-time inference speed.

**Up-Projection and final Flow morph.** After updating $\mathbf{W}_{\text{test}}$, the Morph module uses an *Up-Projection* to map the low-rank embedding $\mathbf{x}_v$ to a $B \times 2$ matrix ($\mathbf{x}_{\text{mod}}$), incurring $\mathcal{O}(d\,B)$ cost. Finally, the Flow layer is applied again with the *morphed* parameters $\mathbf{W}_{\text{flow}} \odot \mathbf{x}_{\text{mod}}$. This second Flow transformation has the same $\mathcal{O}(C\,T\,B)$ complexity discussed in Section L.1.

**Overall cost of Morph.** Hence, for a single batch of length $T$ and channels $C$, test-time training plus the final Flow morph yields:

$$\underbrace{\mathcal{O}(N_{\text{iter}} \times C\,T\,d)}_{\text{self-supervised updates}} + \underbrace{\mathcal{O}(d\,B)}_{\text{up-projection}} + \underbrace{\mathcal{O}(C\,T\,B)}_{\text{Flow morph}}.$$

Since $B$ and $d$ are typically much smaller than $C\,T$ (and since $N_{\text{iter}}$ is kept small), the overall additional time cost of Morph remains reasonable in practical settings.

## L.3. Morph vs. Flow-Only Computation

**Flow-Only Baseline.** Recall that applying the Flow layer to a single batch of length $T$ and $C$ channels costs $\mathcal{O}(C\,T\,B)$ in each forward (or inverse) pass, where $B$ is the number of bins per channel. In a typical forward-only inference setting, if no further transformations are applied, the runtime per batch is dominated by $\mathcal{O}(C\,T\,B)$.

**Added Steps from Morph.** By contrast, when the Morph module is enabled, each batch must go through:

1. The **initial Flow pass** (as in Flow-only): $\mathcal{O}(C\,T\,B)$.

2. **Test-time-trained temporal layer**:
   - Projection to internal dimension $d$ (query/key/value), computed in $\mathcal{O}(C\,T\,d)$.
   - Self-supervised loss and gradient-based updates on $\mathbf{W}_{\text{test}}$, repeated $N_{\text{iter}}$ times for each batch. Each update backpropagates through the projections, adding another $\mathcal{O}(C\,T\,d)$ per iteration.
   - In total, this yields $\mathcal{O}(N_{\text{iter}}\,C\,T\,d)$.

3. **Up-Projection** from dimension $d$ to the bin-height shape $(B \times 2)$, costing $\mathcal{O}(d\,B)$.

4. **Final Flow pass** with morphed parameters, which again is $\mathcal{O}(C\,T\,B)$.

Hence, compared to a Flow-only system, the new steps due to Morph amount to:

$$\underbrace{\mathcal{O}(N_{\text{iter}}\,C\,T\,d)}_{\text{test-time updates}} + \underbrace{\mathcal{O}(d\,B)}_{\text{up-projection}} + \underbrace{\mathcal{O}(C\,T\,B)}_{\text{re-applied Flow pass}},$$

in addition to the baseline Flow cost.

## L.4. Relative Hyper-Parameters Influencing Overhead

- **Bins per channel, $B$.** In Flow-only, the cost scales as $\mathcal{O}(C\,T\,B)$. Morph re-applies the Flow transform once more, increasing the effective factor on $\mathcal{O}(C\,T\,B)$. For small $B$, the overhead from Morph is more pronounced in the self-supervised step; for large $B$, the additional Flow pass also becomes non-trivial.

- **Hidden dimension, $d$.** The internal dimension used in the Morph module's query/key/value projections directly affects the $\mathcal{O}(C\,T\,d)$ cost. A modest $d$ keeps the test-time updates relatively cheap, while a large $d$ can dominate the overall overhead, potentially exceeding the Flow cost if $d \gg B$.

- **Number of iterations, $N_{\text{iter}}$.** Each additional gradient update at test time scales linearly. Even if $d$ is small, large $N_{\text{iter}}$ can inflate the Morph overhead. Balancing between adaptation accuracy and runtime is key.

Empirically, one typically chooses $B$ to be moderate (e.g., $B = 8$ to 16) and $d$ to be significantly smaller than $C \cdot T$, keeping the Morph overhead at a fraction of the total runtime. Furthermore, a small $N_{\text{iter}}$ (often 1–5) can be sufficient to stabilize inference under mild distribution shifts, ensuring that Morph's overhead remains practical compared to Flow-only.

## L.5. Wall-Clock Runtime Comparison

We report average per-batch inference times (in milliseconds per iteration) across six datasets in Table 12. All experiments were conducted using a single NVIDIA Titan RTX GPU paired with an Intel i9-10900KB CPU. Each runtime is measured with input length $T = 96$, batch size $B = 24$, and model dimension $d = 92$.

*Table 12.* Average inference time (ms/iter) for different configurations. The final column reports Morph's share of total MoF runtime.

| Dataset (C) | w/ RevIN | w/ Flow | w/ MoF | Morph in MoF (%) |
|---|---|---|---|---|
| Traffic (C=862) | 107.1 | 115.5 | 122.2 | 44.37% |
| Weather (C=21) | 21.7 | 30.3 | 32.5 | 20.37% |
| ETTh1 (C=7) | 22.1 | 32.5 | 32.9 | 3.70% |
| ETTm1 (C=7) | 23.7 | 32.7 | 33.0 | 3.23% |
| Electricity (C=321) | 42.9 | 48.5 | 51.5 | 34.88% |
| Exchange (C=8) | 22.2 | 32.2 | 32.3 | 0.99% |
| **Average** | – | – | – | **17.92%** |

**Summary.** On average, Morph contributes only **17.92%** of MoF's inference time, and incurs **less than 5%** overhead on total runtime compared to Flow-based baselines. A detailed computational complexity analysis is included in the revised version.

Combining both Flow and Morph, each inference batch involves:

- An initial Flow pass: $\mathcal{O}(C\,T\,B)$.

- Morph test-time update and re-transformation: $\mathcal{O}(N_{\text{iter}}\,C\,T\,d + d\,B + C\,T\,B)$.

While the precise constants depend on implementation details (e.g., bin-search optimization or parallel GPU vectorization), the asymptotic behavior is governed by the linear dependence on $C$, $T$, and $B$, plus the modest overhead of test-time adaptation steps.

# M. Gradient-statistics analysis

For each training step $s$ we collect the gradient tensor $g_s$ of the model parameters and compute:

$$\text{Norm}(s) = \|g_s\|_2,$$

$$\text{Var}(s) = \frac{1}{n}\sum_{i=1}^{n}\left(g_{s,i} - \bar{g}_s\right)^2,$$

$$\text{Skew}(s) = \frac{\frac{1}{n}\sum_{i=1}^{n}(g_{s,i} - \bar{g}_s)^3}{\left(\text{Var}(s)\right)^{3/2}},$$

$$\text{Kurt}(s) = \frac{\frac{1}{n}\sum_{i=1}^{n}(g_{s,i} - \bar{g}_s)^4}{\left(\text{Var}(s)\right)^2} - 3,$$

where $n$ is the number of gradient elements and $\bar{g}_s$ their mean. The kurtosis is reported in its *excess* form (zero for a Gaussian).

The first column of Fig. 23 shows only minor differences among Instance-wise $z$-score, RevIN, and MoF when a shallow Linear backbone is used. In contrast, the second column (PatchTST on ETTh1) reveals visibly smaller gradient norm and variance for MoF, with skewness and kurtosis converging more rapidly. The third column (PatchTST on Exchange) highlights the strongest effect: gradients under MoF remain well-behaved throughout training, reflecting the heavier tails of the dataset and explaining the larger performance improvements.

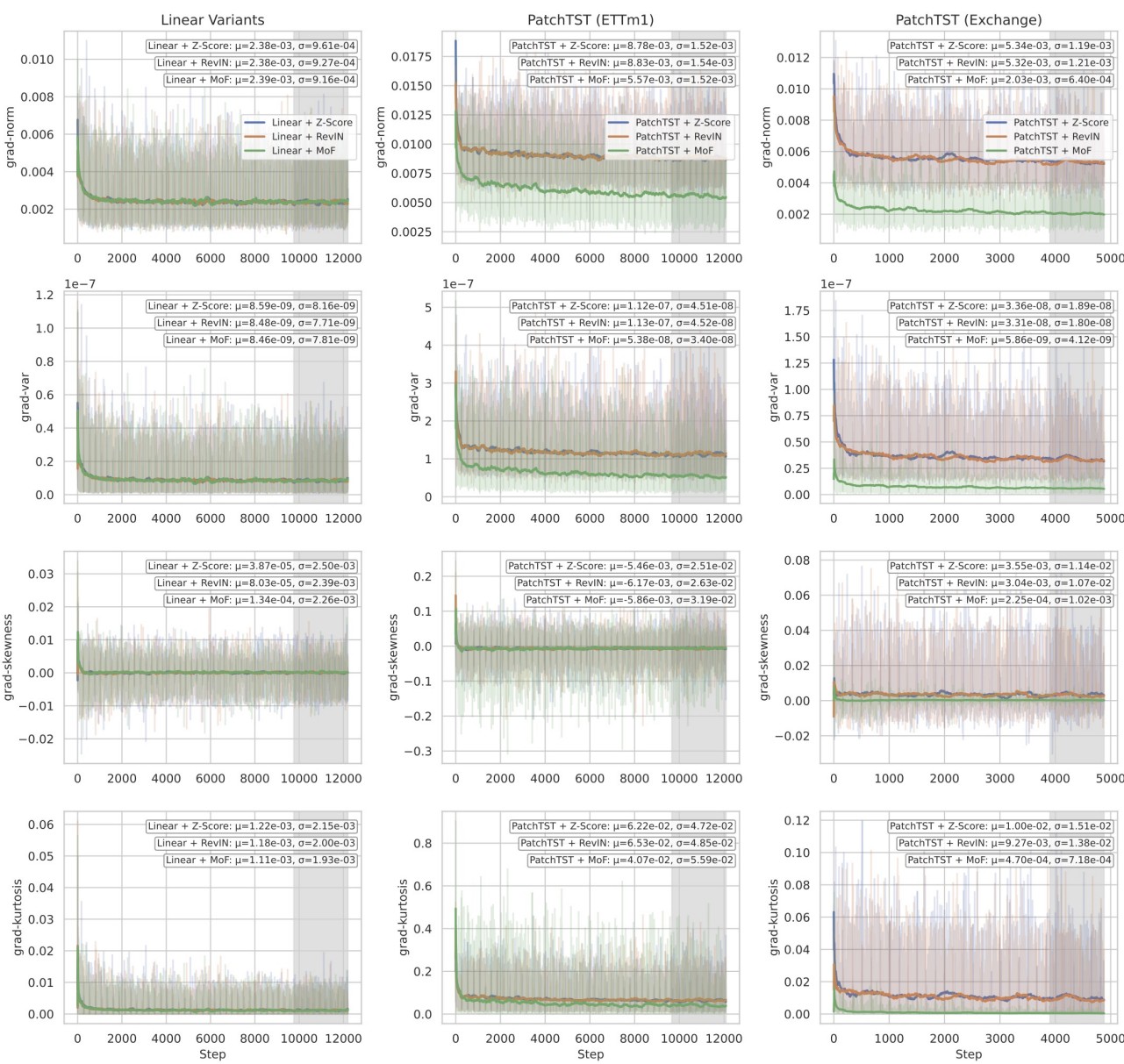

*Figure 23.* **Gradient statistics over training.** Columns: (i) Linear backbone on *Exchange*, (ii) PatchTST on *ETTh1*, (iii) PatchTST on *Exchange*. Rows: gradient norm, variance, skewness, and kurtosis. MoF (green) yields lower norms/variances and markedly smaller higher-order moments—especially on the heavy-tailed *Exchange* dataset—while differences are modest for the shallow Linear model. Shaded areas denote one standard deviation over five runs.

