# OpenReview forum: "Slimming the Fat-Tail: Morphing-Flow for Adaptive Time Series Modeling"
_ICML.cc/2025/Conference — ICML 2025 poster_

### Official Review · Reviewer_sm3N · 2025-03-12

**Overall Recommendation:** 2

**Summary:**

This paper tries to address the challenge of forecasting temporal sequences characterized by non-stationarity and leptokurtic (fat-tailed) distributions. The proposed Morphing-Flow (MoF) framework innovatively integrates a spline-based transform layer (“Flow”) with a test-time-trained adaptation method (“Morph”) to normalize these distributions while preserving essential extremal features. The numerical experiments are conducted.

**Claims And Evidence:**

Since this draft doesn't contain a specific section to directly summarize the major contributions. The evaluations are based reviewer's personal understanding.

1. The methods part

-- The proposed methods are well-motivated.

2. The experiments part

-- After the details results in Table 1 and Appendix E. I obverse significant performance mismatch for benchmarks and their results in literature. For example, in iTransformer original paper the average MSE results are as follows

| Dataset    | Itransformer | PatchTST | Dlinear |
| -------- | ------- | --- | ---|
| ETTh1  | 0.454   |0.469 | 0.456|
| ETTh2 | 0.383     | 0.387| 0.559|
| ETTm1    | 0.407   | 0.387 | 0.403 |
| ETTm2   | 0.288   | 0.281|  0.350|
| ECL    | 0.178   | 0.205 | 0.212 |
|Exchange | 0.360 |  0.367 |  0.354|
| Traffic    | 0.428   | 0.481 |  0.625|
| Weather    | 0.258   | 0.259 | 0.265 |

Can the authors elaborate more on the reason for the mismatch?

**Essential References Not Discussed:**

N/A

**Experimental Designs Or Analyses:**

Please see the above sections.

**Methods And Evaluation Criteria:**

My major concern is the mismatch of the benchmarks' performance in the numerical section. Based on the current presentation, it is hard for me to make fair evaluation.

**Other Comments Or Suggestions:**

N/A

**Other Strengths And Weaknesses:**

N/A

**Questions For Authors:**

At this stage, the aforementioned mismatch issue blocks me from providing a comprehensive evaluation. I will defer my final decision until this concern is adequately addressed.

**Relation To Broader Scientific Literature:**

N/A

**Theoretical Claims:**

N/A

---

> ### Author Rebuttal · Authors · 2025-04-01
>
> Thank you for raising this concern.
>
> ---
>
> > **Question:**
> > *Regarding baseline performances*
>
> Prior works, such as iTransformer and PatchTST, adopt **different hyperparameters for different datasets**, as summarized in the table below. This practice makes it difficult to disentangle performance gains stemming from model design versus hyperparameter tuning—potentially introducing *test-set overfitting*.
>
>
> | Hyper-param      | Ours          | iTransformer        | PatchTST             |
> |--------------------|---------------|----------------------|-----------------------|
> | lookback length    | 336           | 96                   | 336                   |
> | Patch Len/Stride   | 16/8          | 12 / unknown    | 16/8 to 24/2          |
> | d_model            | 512           | 128–512              | 16–128                |
> | d_ff               | 512           | 128–2048             | 128–256               |
> | learning_rate      | 1e-4          | 1e-3 to 5e-5         | 1e2 to 1e-4           |
> | batchsize          | 4             | 16–32                | 8–128                 |
>
> As noted above, baseline performance in prior works often varies due to differing experimental setups.  For low-hyperparameter models like *DLinear*, our results closely match both the original paper and other reimplementations under comparable settings on overlapping datasets.
>
> | *Dlinear(MSE)*    | Ours  | [DLinear](https://arxiv.org/abs/2205.13504) (AAAI 2023) | [LIFT](https://arxiv.org/pdf/2401.17548)(ICLR 2024) | [SAN](https://openreview.net/forum?id=5BqDSw8r5j) (NeuraIPS 2023) |
> |-------------|-------|-----------------------------|-----------------|-------------------|
> | Weather     | 0.245 | 0.246                       | 0.246           | 0.245             |
> | Electricity | 0.167 | 0.166                       | 0.166           | 0.166             |
> | Traffic     | 0.435 | 0.434                       | 0.434           | 0.435             |
>
>
> **Our baseline experiments** use a single, unified hyperparameter configuration across all methods and datasets (including ours; see Appendix C), enabling a *fairer* and more *challenging* comparison by removing dataset-specific tuning.  Even under this stricter setup, our re-implementations of strong baselines (e.g., iTransformer, PatchTST, DLinear) outperform reported results—e.g., those from the iTransformer paper—by **1.69%–19.1%** on average across datasets (see summary table below; full results [here](https://anonymous.4open.science/r/Materials-1D2F/param_cmp.pdf)).
>
> | **Avg. MSE**     |  by us           |    by iTrans.   |       by PatchTST     |
> |--------------------|---------------|----------------------|-----------------------|
> | iTrans             | **0.343 (-2.72%)**| 0.345                | -                     |
> | PatchTST           | **0.349 (-4.35%)**| 0.353                | 0.307                 |
> | DLinear            |**0.326 (-19.1%)**| 0.403                | 0.330                 |
>
>
> **Our proposed MoF module (with Mamba backbone)** still achieves substantial gains, despite these stronger baselines, improving over iTransformer by **14.9%** and over PatchTST by **16.9%** in average MSE.
>
> Thank you for raising this concern. We will emphasize these clarifications in the revised paper to preempt any potential reader confusion.
>
> ---
>
> We hope this addresses your concern and if you have any further questions or concerns, we are happy to address them.

---

### Official Review · Reviewer_fqDp · 2025-03-12

**Overall Recommendation:** 2

**Summary:**

This paper proposes Morphing-Flow, a spline transformation module coupled with test-time adaptation, to counter fat-tailed distributions and distribution shifts.

**Claims And Evidence:**

The paper claimed that fat-tailed distributions have negative effects on model convergence. This claim is supported by some empirical experiments on synthetic data.

**Essential References Not Discussed:**

This paper lacked some discussions and comparisons with more recent papers. For example, [1] is closely related to this work because it also inroduces normalizaiton flow to address distribution shifts; [2] is an efficient and effective time-series model that operates in the frequency domain; [3] is a successor of PatchTST that is capable of delivering forecasts for any horizons using a single model. Both [3] and PatchTST leverage RevIN as the normalizaiton method and seem to work well in practice.

[1] IN-Flow: Instance Normalization Flow for Non-stationary Time Series Forecasting, https://arxiv.org/abs/2401.16777
[2] FITS: Modeling Time Series with 10k Parameters, https://arxiv.org/abs/2307.03756
[3] ElasTST: Towards Robust Varied-Horizon Forecasting with Elastic Time-Series Transformer, https://arxiv.org/abs/2411.01842

**Experimental Designs Or Analyses:**

I have checked the experiments and analyses.

**Methods And Evaluation Criteria:**

Empirical experiments on synthetic data seem to suggest that fat-tailed distributions could be a severe problem affecting model convergence. However, existing time-series approaches seem to work well in practical data, and this paper also introduces test-time adaptation to adjust the parameters of the normalizaiton module. These contradictions made me hard to believe what is the real contribution behind Morphing-Flow.

**Other Comments Or Suggestions:**

I would like to see some concrete and real evidences demonstrating the importance of tackling fat-tailed distributions in time-series forecasting. Current results make me feel that RevIN is already good enough while MoF only makes marginal improvements but at huge costs and with complicated designs.

**Other Strengths And Weaknesses:**

This paper proposes an interesting angle. But current experiments can hardly convince me that fat-tailed distribution is a severe problem in time-series forecasting in real-world scenarios.

**Questions For Authors:**

- I have concerns on the real importance of dealing with fat-tailed distributions in time-series data. Although the synthetic experiments (figure 2) seem to suggest this could be a severe problem, concrete experiments on real-world data, comparing MoF with other normalization methods (Appendix I, page 28) does not show significant performance gains, especially when compared with the simple and effective baseline, RevIN. I would like to understand why the authors claimed this as a severe problem in time-series forecasting and designed such a complicated method to tackle it.

- If the MoF module does not play a criticle role, the remaining contribution of this paper would be test-time adaptation. However, in this context, directly comparing test-time tuned MoF models with traditional time-series models only trained on the training data is not fair. For example, you can also apply test-time adaptation to the parameters of RevIN, which may also help.

- Moreover, if MoF is designed as a model-agnostic module to normalize data distributions, have you tried to combine this with more forecasting models beyond DLinear and Mamba? Table 3 includes similar experiments but using some baselines that have well-known performance issues, such as Informer and Autoformer, in the liteature. It is well-known that PatchTST is equipped with RevIN by default. Have you considered combining the backbone of PatchTST with MoF to check how MoF advance over RevIN?

- Figure 8 compares Linear-MoF with other baselines, but only on ETTh2 dataset. It is well known that on this dataset, the Linear model could be better, but on other complicated scenarios, PatchTST and iTransformer could still significantly improve Linear. Only showing results on one dataset could be very misleading.

- Similarly, Figure 9 compares MoF with other normalization methods, but only on the Weather dataset. After checking Appendix I, I find that MoF does not always ensure the best normalizaiton across diverse datasets. Hence Figure 9 could largely mislead readers, too.

**Relation To Broader Scientific Literature:**

The fat-tailed distribution problem may also exist in other scenarios.

**Theoretical Claims:**

No. This paper does not include theoretical proofs.

---

> ### Author Rebuttal · Authors · 2025-04-01
>
> Thank you for your comments. Please kindly find our response below.
>
> ---
> > **Comment 1:**
> > *Regarding the contribution*
>
> As shown in Fig.1, while stabilizing methods normalize data, they often expose heavier tails(often exceeding those in our synthetic setups) in real datasets—outliers that disrupt gradient flow. Our Flow remaps outliers into a stable range, and Morph adapts to evolving data at test time.
>
> In summary, with *new ablations on PatchTST*, our contributions are:
>
> - **Flow as the Primary Contributor** mitigates outlier-driven gradient spikes and yields a **5.95%** MSE reduction—exceeding the gain from any other individual component. Showing that fat-tail suppression is the principal driver of MoF’s effectiveness.
>
> - **Morph for Distribution Shifts** provides a **3.11%** MSE gain by adapting to distributional shifts via test-time tuning of RevIN’s affine parameters.
>
> - **Flow and Morph work synergistically**: Morph benefits from the stabilized gradient flow, resulting in a **7.17% MSE reduction** over PatchTST.
>
> We revised the introduction to highlight these insights.
>
> |Patch TST +|Average MSE across 8 datasets|-Δ (%)|
> |-|-|-|
> |no Norm|0.370|-4.33%|
> |RevIN|0.349|0.00%|
> |Morph|0.333|3.11%|
> |Flow|0.319|5.95%|
> |Morph+Flow|0.314|**7.17%**|
> \* [[Full Result Here]](https://anonymous.4open.science/r/Materials-1D2F/abla.pdf)
>
> ---
> > **Question 1:**
> > *Regarding the importance of dealing with fat-tailed distributions and performance comparison with Linear Backbone(Appendix I)*
>
> Real data (finance [1], climate [2], health [3]) often has frequent outliers that dominate loss and disrupt training dynamics. The *Linear* backbone used in Appendix I often underfits, hiding these effects (Flow and RevIN both show limited gains).
>
> Added gradient analysis ([link](https://anonymous.4open.science/r/Materials-1D2F/gradient_stats_grid.pdf)) shows MoF reduces skewness/kurtosis for smoother gradients, crucial in larger model & fat-tailed data (ETT -> Exchange).
>
> Across datasets, MoF yields +15.0% with PatchTST vs +2.75% with Linear, indicate that more powerful backbones magnify the impact of outliers and thus benefit more from Flow-based normalization.
>
> |Avg. of avg.(MSE)|+RevIN|+MoF|
> |-|-|-|
> |Linear|+1.83%|+2.75%|
> |PatchTST|+5.65%|+15.0%|
>
> [1].Fat tails in leading indicators(Econ Lett 2020)
>
> [2].Emergence of heavy tails in streamflow distributions: the role of spatial rainfall variability (Adv Water Res 2023)
>
> [3].Evidence that coronavirus superspreading is fat-tailed(PNAS 2020)
>
> ---
> > **Question 2:**
> > *Regarding the fariness of test-time training*
>
> As suggested, We updated RevIN’s affine param via Morph(TTT) and achieved **+3.11%** gain(RevIN ± Morph). Meanwhile, Flow (no TTT) gives **+5.95%**, and **Flow+Morph** reaches **+7.17%**, showing that Flow is the key driver, with Morph benefiting stable grad. Our TTT starts from a fixed $W_0$ per instance and modifies only $W$, ensuring no future leakage.
>
> ---
> > **Question 3:**
> > *Have you considered **combining the backbone of PatchTST with MoF** to check how MoF advance over RevIN?*
>
> Yes. PatchTST+MoF improves MSE by 7.17% over PatchTST+RevIN across 8 datasets (wins 7/8). [[Results]](https://anonymous.4open.science/r/Materials-1D2F/patch_mof.pdf) to be added to Table 3 & Appendix J.
>
> ---
> > **Question4:**
> > *Regarding Figure 8 limited to ETTh2 dataset*
>
> Figure 8 shows non-monotonic input-length effects on given dataset (Linear-MoF peaks at 192, iTransformer at 336), *not* meant for overall ranking. More results in Appendix H.
>
> In fact, our gains aren’t ETTh2-specific, Table 1 shows simple MoF+Linear outperforms iTransformer by 7.5% and PatchTST by 9.2% across 8 sets, trailing only on ETTm2. Stronger backbones can further amplify gains.
>
> We clarified this intent in the updated version.
>
> ---
> > **Question 5:**
> > *Regarding Figure 9 limited to the Weather dataset*
>
> Thank you for pointing this out.
>
> Fig.9, like Fig.8, shows how input length interacts with normalization on one dataset, not a universal ranking. We replaced it with PatchTST-based results, showing MoF’s consistency at larger scale, avoiding future confusion.
>
> ---
> > *Regarding References Not Discussed*
>
> We have added the suggested works to our related section.
>
> *IN-Flow*[1] addresses nonstationarity(not fat-tail) via entangled flows, which can be complex for high-dimensional data. *FITS*[2] is parameter-efficient but needs hyperparameter grid search, which are rather costly to train. *ElasTST*[3] focuses on backbone design for varied-horizon forecasting, orthogonal to our normalization approach. Implementing [1] (no public code) worked for small data but was unstable on high-dim benchmarks. [2] was erratic with our unified hyperparams. [[Full Result]](https://anonymous.4open.science/r/Materials-1D2F/ref_cmp.pdf)
>
> We will include these findings in our revision.
>
> ---
> We hope these address your concerns satisfactorily. If you have any further questions or concerns, we are happy to address them.

---

### Official Review · Reviewer_E595 · 2025-03-13

**Overall Recommendation:** 3

**Summary:**

The work introduces Morphing-Flow (MoF), a framework to address the challenges of fat-tailed distributions in time series forecasting through adaptive normalization and test-time adaptation. MoF combines a spline-based Flow layer for distribution normalization and a Morph module for dynamic adaptation, achieving state-of-the-art performance across multiple datasets. The framework is efficient, robust to hyperparameters, and can be easily integrated into various models, offering a practical solution for improving forecasting accuracy in non-stationary environments.  MoF achieves state-of-the-art performance on multiple benchmark datasets, outperforming existing models by an average of 6.3%, and operates efficiently with a simple linear backbone, achieving comparable performance to complex models while using significantly fewer parameters.

**Claims And Evidence:**

I think most claims that are made in the paper are supported by clear evidence. For example,

1. The authors demonstrate the effectiveness of MoF in reducing fat-tailed distributions through strong experiments MoF significantly reduces excess kurtosis (a measure of fat-tailed distributions) in the transformed data compared to raw or stationarized data. This is supported by visualizations and quantitative results across multiple datasets (e.g., ETTh2, ETTm1, Electricity, Weather).

2. The authors demonstrate that MoF is plug and play through experiments with different model architectures (Autoformer, Informer and DLINEAR).

**Essential References Not Discussed:**

I didn't find essential references that are not discussed.

**Experimental Designs Or Analyses:**

Yes I checked the soundness of experimental designs, from datasets, metrics, baselines to the experiment results. I didn't find apparent issues.

**Methods And Evaluation Criteria:**

The proposed methods and evaluation criteria in the paper "Slimming the Fat-Tail: Morphing-Flow for Adaptive Time Series Modeling" are overall aligned with the problem of handling non-stationary, fat-tailed distributions in time series forecasting.

For the proposed Flow Layer (Spline-Based Transformation), it addresses a critical issue in time series forecasting—fat-tailed distributions that destabilize model training and prediction. By using a spline-based transformation to normalize these distributions, the Flow layer directly targets the problem of high kurtosis and skewness.

For the proposed Morph Module, it tackles distribution shifts between training and testing data, which is particularly important in real-world applications where data characteristics can change over time.

For the metrics, the authors use MSE and MAE, which are standard for evaluating time series forecasting models and provide a clear measure of prediction accuracy. By using both MSE and MAE, the authors capture different aspects of model performance (sensitivity to outliers and overall error magnitude).

**Other Comments Or Suggestions:**

Section 2: The term "excess kurtosis" is used without defining it. Adding a brief definition or reference to the appendix would be helpful for readers unfamiliar with the term.

**Other Strengths And Weaknesses:**

The paper is overall written with clear logic.

**Questions For Authors:**

1. Can the authors provide more details on the computational overhead introduced by the Morph module during inference? Specifically, how does the Morph module's test-time adaptation affect the runtime performance compared to models without this adaptation?

2. The Flow layer uses a spline-based transformation to normalize fat-tailed distributions. Can the authors provide more interpretability analysis or case studies to explain how the spline transformation affects specific features or time series patterns?

**Relation To Broader Scientific Literature:**

The MoF framework directly addresses the challenge of fat-tailed distributions by using a spline-based transformation (Flow layer) to normalize the data. This approach is novel in the context of neural network-based time series forecasting and provides a structured way to mitigate the impact of fat-tailed noise on model convergence and performance. MoF can integrate with various backbones, including Transformer-based and Mamba-based architectures.

**Theoretical Claims:**

No theoretical claims.

---

> ### Author Rebuttal · Authors · 2025-04-01
>
> Thank you for your feedback!
>
> ---
>
> > **Suggestions 1:**
> > *Section 2: The term "excess kurtosis" is used without defining it.*
>
> Thank you for the helpful comment.
>
> A definition of *"excess kurtosis"* was included in Appendix C.2: it measures the tailedness of a distribution relative to a Gaussian distribution, which has zero excess kurtosis.
>
> We've revised the wording and added a pointer from the main text to improve accessibility.
>
> ---
>
> > **Questions 1:**
> > *Can the authors provide more details on the **computational overhead introduced by the Morph module** during inference? Specifically, how does the Morph module's test-time adaptation affect the runtime performance compared to models without this adaptation?*
>
> Thanks for the question.
>
> The Morph module introduces test-time adaptation overhead, which scales as:
>
> ```
> O(N_iter × C × T × d)  # test-time gradient updates
> + O(d × B)             # up-projection
> + O(C × T × B)         # re-applied Flow pass
> ```
>
> With small projection dimension `d` in Morph, a moderate number of bins `B` in the Flow, and a fixed, small number of test-time iterations `N_iter`, the overall asymptotic complexity remains dominated by the shared Flow component, i.e., `O(C × T × B)`.
>
> We report average per-batch inference times (ms/iter) across six datasets:
>
> | _Runtime(ms/iter)_ | **w/ RevIN** | **w/ Flow** | **w/ MoF** | **Runtime of Mo in Mof(%)** |
> |--------------------|--------------|------------------|------------|---------------------|
> | Traffic(C=862)     | 107.1        | 115.5            | 122.2      | 44.37%              |
> | weather(C=21)      | 21.7         | 30.3             | 32.5       | 20.37%              |
> | ETTh1(C=7)         | 22.1         | 32.5             | 32.9       | 3.70%               |
> | ETTm1(C=7)         | 23.7         | 32.7             | 33         | 3.23%               |
> | electricity(C=321) | 42.9         | 48.5             | 51.5       | 34.88%              |
> | exchange(C=8)      | 22.2         | 32.2             | 32.3       | 0.99%               |
> |                    |              |                  |            | **17.92%**          |
> *Tested with T=96, B=24, d=92*
>
> On average, Morph accounts for ~17.92% of MoF’s inference time, with less than 5% overhead on overall model runtime.
>
>
> We’ve included this in the revised version with detailed complexity breakdowns.
>
> ---
>
> > **Question 2:**
> > *The Flow layer uses a spline-based transformation to normalize fat-tailed distributions. Can the authors provide **more interpretability analysis or case studies** to explain how the spline transformation affects specific features or time series patterns?*
>
> Thank you for the suggestion.
>
> Our Flow layer applies a monotonic and differentiable spline transformation to **re-map frequent outliers into a more regular, Gaussian-like range**.  This normalization compresses extreme values, re-centering the effective support into the region where activations and gradients are more stable. The result is improved numerical conditioning during optimization.
>
> We added two new visual components:
> - A case study visualizing how Flow attenuates heavy-tailed spikes while preserving underlying temporal structure ([Illustration](https://anonymous.4open.science/r/Materials-1D2F/illus.pdf)).
> - A gradient dynamics study demonstrating that Flow stabilizes backpropagation (reduced norm, skewness, and kurtosis), detailed in ([Gradient Statistic Dynamics](https://anonymous.4open.science/r/Materials-1D2F/gradient_stats_grid.pdf)).
>
> In **Figure 4**, subfigure (2) shows a heavy-tailed (green) distribution from ETTh2, which was transformed by Flow (subfigure (3)) into a more symmetric, near-Gaussian form (blue, subfigure (4)). These results are now further contextualized in the revised text with a dedicated paragraph discussing how Flow impacts both distribution geometry and optimization dynamics.
>
> ---
>
> We hope these address your concerns satisfactorily. If you have any further questions or concerns, we are happy to address them.

---

### Decision · Program_Chairs · 2025-05-01

**Decision:**

Accept (poster)

**Comment:**

This submission proposes Morphing-Flow, a two-component normalization framework that (i) uses a spline-based Flow layer to tame fat-tailed inputs and (ii) applies a light test-time Morph update to track distribution drift.

Strengths
- Clearly motivated attack on the under-studied “fat-tail” failure mode; Flow alone already yields sizable gains.
- unified tuning, ablations isolating Flow vs. Morph, new PatchTST+MoF results added in the rebuttal, and detailed runtime analysis (Morph ≈ 18 % of inference cost).
- Plug-and-play design that fits multiple backbones without architectural changes.

Weaknesses / how addressed
- authors showed that earlier papers tune per-dataset, whereas their single-config evaluation is stricter; recreated baselines actually perform better than literature, yet MoF still wins.
- confusion came from an appendix using a deliberately weak Linear backbone. PatchTST+MoF gains (+7.2 % MSE, wins 7/8 datasets) clarify impact.

I vote for a weak accept.